# Does Your Reasoning Model Implicitly Know When to Stop Thinking?

**Zixuan Huang** [1 2]  **Xin Xia** [2]  **Yuxi Ren** [2]  **Jianbin Zheng** [2]  **Xuanda Wang** [2]  **Zhixia Zhang** [1 *]  **Hongyan Xie** [1]
**Songshi Liang** [2]  **Zehao Chen** [1]  **Xuefeng Xiao** [2]  **Fuzhen Zhuang** [3]  **Jianxin Li** [1]  **Deqing Wang** [1]  **Yikun Ban** [1 †]

## Abstract

Recent advancements in large reasoning models (LRMs) have greatly improved their capabilities on complex reasoning tasks through long Chains of Thought (CoTs). However, this approach often results in substantial redundancy, impairing computational efficiency and causing significant delays in real-time applications. Recent studies show that longer reasoning chains are frequently uncorrelated with correctness and can even be detrimental to accuracy. In a further in-depth analysis of this phenomenon, we surprisingly uncover and empirically verify that LRMs implicitly know the appropriate time to stop thinking, while this capability is obscured by current sampling paradigms. Motivated by this, we introduce **SAGE** (**S**elf-**A**ware **G**uided **E**fficient Reasoning), a novel sampling paradigm that unleashes this efficient reasoning potential. Furthermore, integrating **SAGE** as mixed sampling into group-based reinforcement learning (**SAGE-RL**) effectively incorporates SAGE-discovered efficient reasoning patterns into standard pass@1 inference, markedly enhancing both the reasoning accuracy and efficiency of LRMs across multiple challenging mathematical benchmarks.

## 1. Introduction

Reinforcement learning from verifiable rewards (RLVR) algorithms, such as GRPO (Shao et al., 2024; Yang et al., 2026) and GSPO (Zheng et al., 2025a; Lu et al., 2026), have played a pivotal role in enabling test-time scaling. This capability allows large reasoning models (LRMs) like o3 (OpenAI, 2025a) and DeepSeek-R1 (Guo et al., 2025) to

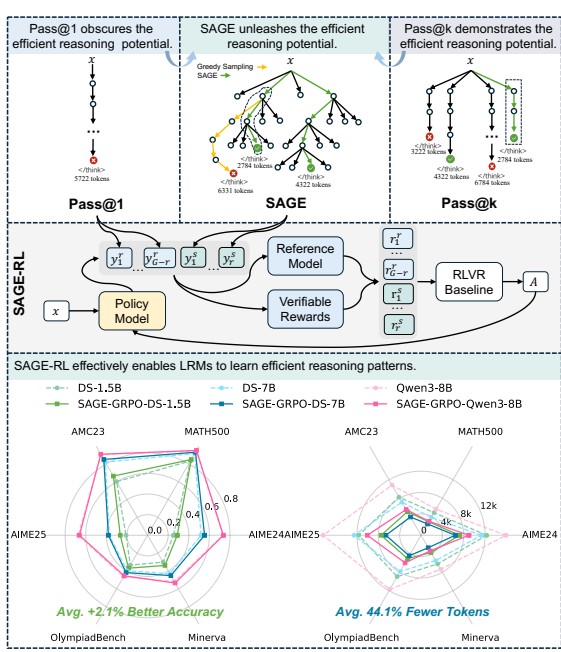

*Figure 1.* SAGE unleashes the efficient reasoning potential of LRMs obscured by pass@1 and identifies the optimal completions within the model's capability hidden in pass@k. By enabling LRMs to learn these efficient reasoning patterns, SAGE-RL-tuned models simultaneously enhance reasoning capacity and conciseness on multiple challenging mathematical benchmarks.

"think longer". Longer CoTs enable LRMs to explore intermediate steps in greater depth and reduce abrupt logical leaps, thereby achieving unprecedented performance on challenging reasoning benchmarks such as AIME (Art of Problem Solving, 2024), OlympiadBench (Chaoqun et al., 2024) and IMO (Luong et al., 2025).

While longer reasoning chains are expected for solving harder problems, prior work shows that length inflation can be uncorrelated with correctness, and that *shorter* chains may in fact yield *better* accuracy. For example, Balachandran et al. (2025) observe that on AIME 2025, DeepSeek-R1 produces responses nearly 5× longer than Claude 3.7 Sonnet while achieving comparable accuracy; Hassid et al. (2025) show that on AIME and HMMT, the shortest responses from QwQ-32B outperform randomly sampled ones by 2 percentage points using 31% fewer tokens. These findings collectively reveal that current CoT outputs often contain

**For implement details, contact** huang_zx@buaa.edu.cn.
[*]Second Student Author. [1]Beihang University [2]Bytedance China [3]School of Artificial Intelligence, Beihang University. Correspondence to: Yikun Ban <yikunb@buaa.edu.cn>.

*Proceedings of the 43^{rd} International Conference on Machine Learning*, Seoul, South Korea. PMLR 306, 2026. Copyright 2026 by the author(s).

substantial redundancy and irrelevant tokens that do not contribute to the final solution. These unnecessary tokens dramatically reduce reasoning efficiency. This naturally raises a pertinent question: *do LRMs know the appropriate time to terminate thinking?*

We find that, during the exploration of multiple reasoning chains, LRMs consistently assign high confidence to concise yet effective reasoning paths. However, current sampling-based inference strategies typically overlook or fail to select these short and effective chains. Moreover, this phenomenon exhibits clear convergence behavior and becomes increasingly pronounced as the exploration space expands. Taken together, these results strongly indicate that **reasoning models implicitly know the appropriate moment to terminate their reasoning process, but this capability is obscured by current pass@1 training and inference paradigms**.

Motivated by this insight, we introduce **SAGE** (**S**elf-**A**ware **G**uided **E**fficient Reasoning), a simple yet effective decoding strategy that leverages the reasoning model's self-confidence to discover relatively precise reasoning chains. By incorporating **SAGE** as mixed sampling into group-based reinforcement learning (**SAGE-RL**), we enable the reasoning model to learn concise yet effective thinking patterns without altering its original reasoning paradigm.

In summary, our contributions in this work are as follows:

- We uncover and demonstrate that **LRMs implicitly know the appropriate time to stop thinking**, but this capability is obscured by current sampling paradigms.

- We propose **SAGE**, a novel sampling paradigm that unleashes the efficient reasoning potential of LRMs, simultaneously improving both accuracy and conciseness of reasoning chains.

- We propose **SAGE-RL**, a simple modification to RLVR frameworks that integrates SAGE into the rollout process. As shown in Figure 1, SAGE-RL-tuned models achieve consistent gains across six challenging reasoning benchmarks, including MATH-500, AIME 2024, AIME 2025, AMC23, OlympiaBench and Minerva.

## 2. Dilemmas of Reasoning Models under Current Sampling Paradigms

To investigate whether reasoning models possess the ability to recognize the appropriate moment to terminate thinking, we first need to re-examine the dilemmas faced by these models under current sampling paradigms.

**Pass@k: Scaling CoT length does not lead to correct answers.** Assuming that LRMs using current sampling paradigms can reliably stop thinking at the appropriate moment, longer CoTs should outperform shorter ones in leading to correct solutions. However, extensive experiments involving multiple samplings of the same problem refute this assumption. Balachandran et al. (2025) observes that on AIME 2025, DeepSeek-R1 produces responses nearly 5× longer than Claude 3.7 Sonnet while achieving comparable accuracy; Hassid et al. (2025) also shows that on AIME and HMMT, the shortest responses from QwQ-32B outperform randomly sampled ones by 2 percentage points using 31% fewer tokens. Shrivastava et al. (2025) found that, on AIME 2025, in 72% of problems where both correct and incorrect answers were generated, the longer response was more likely to be incorrect than the shorter one.

These findings collectively reveal that: once the chain-of-thought length reaches a certain threshold, simply scaling the length further does not lead to a corresponding improvement in the model's reasoning capability. *Furthermore, the optimal response within the model's capability is obscured by existing sampling paradigms and can currently only be retrieved post hoc through test-time scaling methods.*

**Pass@1: Existing sampling strategies fail to enable timely termination of thinking.** To gain a finer-grained understanding of these findings and precisely locate its root cause, we build upon this observation and take a step further. Reasoning tasks, particularly in mathematical reasoning and code generation, typically require step-by-step answers. Leveraging this observation, we introduce a simple metric to quantify the efficient reasoning capability of models: the **Ratio of the First Correct Step** (RFCS), defined as the step index at which the correct answer first appears divided by the total number of reasoning steps.

Specifically, we utilize DeepSeek-distilled-Qwen-1.5B (DS-1.5B), (DeepSeek-AI, 2025), DeepScaleR (Luo et al., 2025b) and Qwen3-8B (Yang et al., 2025a) to generate answers for MATH-500 (Lightman et al., 2023) problems. For each response, we segment it into distinct reasoning steps by "\n\n" (Chen et al., 2025a) and compute RFCS for each problem. As illustrated in Figure 2, the model correctly derives the answer using only 500 tokens, yet under the current sampling strategy, it continues with an additional 452 redundant tokens before terminating the reasoning process. This clearly demonstrates that the LRM fails to end its thinking at the appropriate moment on this problem.

Such cases are not isolated in our study. More statistical results are summarized in Figure 3, where **RFCS(< 1)** and **RFCS(avg)** respectively denote the number of correct responses where RFCS is not equal to 1 and the average RFCS value across all correct responses. From the statistical results, all models exhibit significant ineffective steps in over half of the samples. Moreover, compared to DS-1.5B, models with higher post-training extent (DeepScaleR), or more advanced reasoning capabilities (Qwen3-8B) show no substantial improvement on this metric. This indicates that,

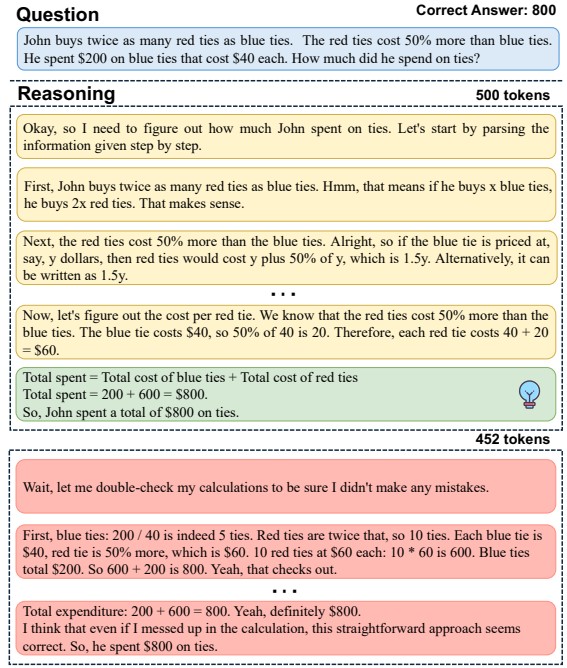

*Figure 2.* Illustration of the step-by-step answering process.

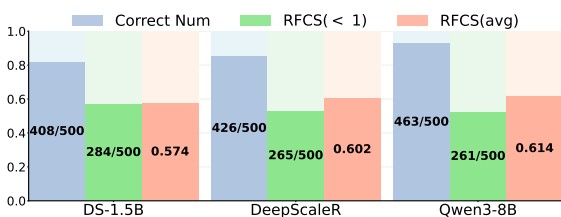

*Figure 3.* Statistics of RFCS on MATH500 across LRMs.

in general scenarios, existing reasoning models struggle to terminate their thinking process at the appropriate moment under the current inference paradigm (i.e., pass@1).

**In summary, the surprising performance of relatively shorter responses in pass@k reveals the inherent potential of the model for efficient reasoning. The pervasive redundancy of reasoning steps in pass@1 indicates that current sampling paradigms obscure this potential.**

Therefore, we attempt to adopt a sampling strategy with a larger exploration space built upon pass@1 to intentionally uncover the precise reasoning chains that are hidden within the broader pass@k distribution.

## 3. Intentionally Exploring Shorter CoTs

**Notations.** Given a query $\mathbf{x}$ and a prefix $\mathbf{y}_{<k} = (y_1, y_2, ...y_{k-1})$ previously generated by the language model $\pi_\theta$, We define $\Phi$ as the average cumulative log-probability up to generation step $k$, where $\phi(y_i; \mathbf{y}_{<i})$ is

the (next-token) log-probability of the $i$-th token in $\pi_\theta$:

$$\Phi(\mathbf{y}_{\leq k}) = \frac{1}{k} \sum_{i=1}^{k} \phi(y_i; \mathbf{y}_{<i}). \qquad (1)$$

$$\phi(y_i; \mathbf{y}_{<i}) = \log \pi_\theta(y_i \mid \mathbf{y}_{<i}, \mathbf{x}). \qquad (2)$$

**Token-Wise Reasoning Path Exploration.** We first propose a token-wise reasoning path expansion algorithm until a maximum step budget $T_{\max}$ is reached. With exploration width $m$ (denoted as **EW**), we maintain the top-$m$ candidate sequences according to the scoring function $\Phi$, and expands them in subsequent decoding steps (Meister et al., 2020).

Formally, given a set of $m$ candidate sequences $Y_{i-1} = \{\mathbf{y}_{\leq i-1}^{(1)}, \mathbf{y}_{\leq i-1}^{(2)}, \ldots, \mathbf{y}_{\leq i-1}^{(m)}\}$ at timestep $i-1$, for $\mathbf{y}_{\leq i-1}^{(j)} \in Y_{i-1}, j \in [1, m]$, we select the top $2m$ most probable tokens

$$\mathcal{T}^{(j)} = \text{Top}_{2m}\left(\{y_i \mid y_i \in \mathcal{V}\}; \phi(\cdot; \mathbf{y}_{\leq i-1}^{(j)})\right) \qquad (3)$$

where $\text{Top}_m(\cdot; \phi)$ denotes an operator that ranks all candidate elements in descending order according to their $\phi$-scores and returns the subset consisting of the top $2m$ elements with the highest scores. This yields a candidate group of size $2m \times m = 2m^2$, formally written as

$$\hat{Y}_i = \{\mathbf{y}_{\leq i}^{(j,k)} \mid j \in [m], k \in [2m]\}, \qquad (4)$$

where each candidate sequence is constructed by appending the $k$-th best token to the $j$-th beam:

$$\mathbf{y}_{\leq i}^{(j,k)} = \mathbf{y}_{\leq i-1}^{(j)} \oplus y_i^{(j,k)}, \quad y_i^{(j,k)} \in \mathcal{T}^{(j)}. \qquad (5)$$

We retain the top-$m$ highest-scoring candidate sequences for next iteration:

$$Y_i = \text{Top}_m\left(\{\mathbf{y}_{\leq i}^{(j,k)} \mid j \in [m], k \in [2m]\}; \Phi\right). \qquad (6)$$

**Exploration Termination.** We denote the **T**olerance accept rank **R**atio of </think> $h/2m$ as **TR**, where $h \in \{1, 2, \ldots, 2m\}$ is a hyperparameter representing the tolerance for the rank of </think> . Given the required number of CoTs $r \in \{1, 2, \ldots, m\}$, once we have reached a candidate sequence $\mathbf{y}_{\leq i}^{(j,k)}$, where $y_i^{(j,k)}$ is </think> and within the top-$h$ probable tokens $\text{top}_h \mathcal{T}^{(j)}$, we add it as a completion to the candidate sequence set $\mathcal{O}$. Otherwise, we discard this candidate sequence, as the model's confidence in terminating the thinking process is low at this point (Liu et al., 2025). When $|\mathcal{O}| \geq r$, we terminate the entire process. If when $i = T_{max}$ and $|\mathcal{O}| < r$, we also add $\text{top}_{r-|\mathcal{O}|}\left(\{Y_{T_{max}} \oplus </think>\}; \Phi\right)$ to $\mathcal{O}$ to ensure $|\mathcal{O}| = r$.

**Greedy Sampling of the Answers.** Through the above process, we generate $r$ reasoning chains $\mathbf{t}_i \in \mathcal{O}$ for each question $\mathbf{x}$. Next, we derive the answer $\pi_\theta(\mathbf{a}_i|\mathbf{x}, \mathbf{t}_i)$ greedily based on the query and the internal reasoning chains. Ultimately, for each question $\mathbf{x}$, our decoding strategy generates $r$ completions $\{\mathbf{t}_i, \mathbf{a}_i\}, i \in [r]$.

Notably, although our algorithm is built upon vanilla beam search, it exhibits significant differences. We provide a detailed comparative analysis in Appendix B.

## 4. Your Reasoning Model Implicitly Knows When to Stop Thinking

Built upon the observations of Section 2 and take a step further, we conduct analytical experiments involving the following related algorithms:

**TSearch(m, r) w/ $\Phi$** denotes the algorithm from Section 3 with exploration width $m$ and $r$ returned completions.

**TSearch (m, r) w/ $\phi$** is a TSearch variant used to ablate the role of $\Phi$, which greedily retains the top-$m$ candidate sequences with the most probable new token at each step according to the following equation instead of Equation 6:

$$Y_i = \left\{ \mathbf{y}_{\leq i}^{(j,k)} \,\middle|\, (j,k) \in \right.$$
$$\left. \arg \mathrm{Top}_m \left( \{ y_i^{(j,k)} | j \in [m], k \in [2m] \}; \phi(\cdot; \mathbf{y}_{\leq i-1}^{(j)}) \right) \right\}. \quad (7)$$

**EW = 0** denotes greedy sampling, which essentially represents a degeneration of TSearch with no exploration. Li et al. (2023) found that its performance is comparable to the average results obtained from random sampling.

**Random** refers to standard random sampling with temperature and top-p both set to 1.0.

### 4.1. High-Confidence Paths Lead to Efficient Reasoning

We use reasoning chains retained by $\Phi$ to represent the high-confidence paths generated during TSearch. To assess the role of the $\Phi$ during this process, we compare TSearch w/ $\Phi$ with TSearch w/ $\phi$ across increasing exploration width $m$.

Enlarging the exploration width $m$ influences TSearch in two contrasting ways. On the positive side, a broader candidate token window $\mathcal{T}$ facilitates the discovery of more varied reasoning paths and improves the probability of identifying optimal solutions among pass@k samples (Shrivastava et al., 2025; Hassid et al., 2025). On the negative side, a larger $\mathcal{T}$ is used for </think> detection. With **TR** = 1, termination occurs immediately upon </think>'s appearance, probably leading to significant length collapse. The results in Figure 4 clearly demonstrate the pivotal role of $\Phi$.

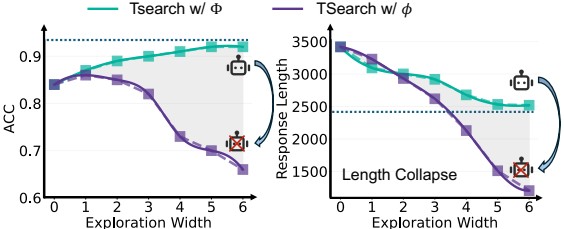

*Figure 4.* Comparison of TSearch variants with increasing **EW** on DS-7B and a randomly selected subset of MATH-500 (size = 100) under a 10k token budget. To directly investigate the influence of $\Phi$, we uniformly set **TR** = 1.

> **Observation 1 (Figure 4).** In TSearch w/ $\Phi$, increasing $m$ leads to a consistent reduction in response length accompanied by a steady improvement in accuracy. By contrast, TSearch w/ $\phi$ suffers a rapid degradation in accuracy that closely tracks the sharp decline in response length. Furthermore, enlarging the exploration space represents an opportunity to enhance reasoning chain quality when $\Phi$ is present, whereas its absence makes length collapse and performance deterioration an inevitable consequence. These results indicate that **the high-confidence branches preserved by $\Phi$ are not only markedly shorter, but also substantially more effective.**

### 4.2. High-Confidence Paths Lead to Confident Ends

To further investigate the length collapse problem in TSearch w/ $\phi$ illustrated in Section 4.1, we apply **TR** to drop branches concluded with low confidence. The experimental results are shown in Table 1.

*Table 1.* Comparison of TSearch (4,1) variants under different **TR** with the same settings of Figure 4. When **TR** < 1, TSearch prunes candidate sequences where the rank ratio of </think> within $\mathcal{T}$ is lower than **TR**. **ACC** denotes the accuracy, **LEN** refers to the average response length, **T-LEN** represents the average number of think tokens.

| Method | TR | ACC | T-LEN | LEN |
|---|---|---|---|---|
| Random | - | 0.84 | 3126 | 3419 |
| TSearch (4,1) w/ $\phi$ | 1.00 | 0.79 | 1712 | 2129 |
| TSearch (4,1) w/ $\phi$ | 0.75 | 0.82 | 2022 | 2333 |
| TSearch (4,1) w/ $\phi$ | 0.50 | 0.89 | 2176 | 2609 |
| TSearch (4,1) w/ $\Phi$ | 1.00 | 0.92 | 2213 | 2609 |
| TSearch (4,1) w/ $\Phi$ | 0.75 | 0.92 | 2221 | 2621 |
| TSearch (4,1) w/ $\Phi$ | 0.50 | 0.91 | 2212 | 2632 |

As for TSearch w/ $\Phi$, varying the **TR** has virtually no impact on performance. By comparison, it exerts a strong influence on TSearch w/ $\phi$. This indicates a strong correlation between the presence of $\Phi$ and the ranking of </think> within $\mathcal{T}$. To further study the correlation between them, we record

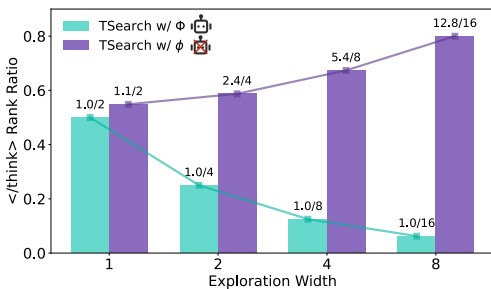

*Figure 5.* The average rank ratio of </think> in $\mathcal{T}$ upon appearance.

the average rank ratio at which </think> appears during TSearch and illustrate them in in Figure 5.

We observe that as **EW** increases, the </think> token identified by TSearch w/ $\Phi$ consistently ranks **first** within the candidate set $\mathcal{T}$ at the moment it appears when evaluated by $\Phi$. This behavior indicates that the policy is highly confident in terminating the reasoning process once </think> enters $\mathcal{T}$. In contrast, for TSearch w/ $\phi$, the rank ratio of the </think> token gradually increases as measured by $\phi$, suggesting increasing uncertainty about whether the next token should be </think>. This discrepancy explains the significant differences in the role of **TR** between TSearch w/ $\Phi$ and TSearch w/ $\phi$, as reported in Table 1.

> **Observation 2 (Figure 6).** The policy implicitly exhibits high confidence in terminating a high-confidence reasoning chain, as supported by TSEARCH with the cumulative probability $\Phi$. However, the final </think> token may have a relatively low next-token probability, which is revealed by TSEARCH w/ $\phi$. This discrepancy indicates that many short yet high-quality reasoning chains are likely to be overlooked by greedy or random sampling strategies.

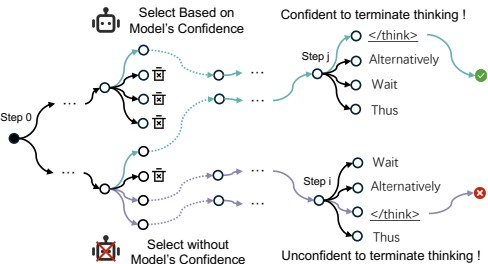

*Figure 6.* Illustration of Observation 2. When reasoning branches are retained according to the model's confidence at each expansion step, the model is able to conclude them with strong confidence.

### 4.3. Scaling Exploration Drives Capability Convergence

In this section, we conduct further experiments to probe the upper boundary of the efficient reasoning capability illustrated in Section 4.1. Specifically, under sufficient token

budget $T_{\max}$ = 32,768, we adopt TSearch (m, 1) w/ $\Phi$ as the sampling strategy, and compare the pass@1 and response length of DS-1.5B and DeepScaleR on MATH-500 and AMC23 as the exploration width $m$ increases. An increase in $m$ corresponds to a larger exploration space during the generation of reasoning chains. The results are shown in Figure 18. For a clearer visualization of the model's performance trends, we measure reasoning efficiency for each run in Figure 18 using token efficiency (pass@1 / response length), as illustrated in Figure 7.

**(1)** At **EW** = 0, the model operates in a completely non-exploratory regime and exhibits limited reasoning efficiency. This indicates that standard non-exploratory greedy or random sampling constrains the model's inherent ability, which is fully consistent with the observations in Section 2.

**(2)** As shown in Figure 18, enlarging the exploration width leads to consistent improvements in pass@1 while simultaneously reducing response length, with both metrics exhibiting a trend toward gradual convergence. This trend further verifies reasoning models' inherent efficient reasoning capability. From Figure 7, we can clearly find that this capability is progressively unleashed as the exploration width grows.

**(3)** LRMs gradually approach the boundary of their inherent efficient reasoning capability as the degree of exploration increases, and this phenomenon is not an isolated occurrence but a universal pattern observed across models and datasets.

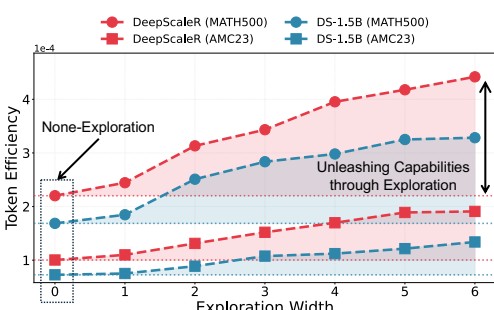

*Figure 7.* Token efficiency comparison on each run in Figure 18.

> **Observation 3 (Figure 7).** As the exploration space expands during reasoning, LRM is increasingly capable of identifying precise and compact reasoning paths with high confidence. Furthermore, with the continued growth of the exploration space, this behavior demonstrates an obvious convergence trend.

Furthermore, as a post-trained version of DS-1.5B, DeepScaleR exhibits steeper token efficiency improvement on both MATH-500 and AMC23. This suggests that greater post-training enhances the model's ability to leverage increased exploration space for unleashing its intrinsic efficient reasoning potential.

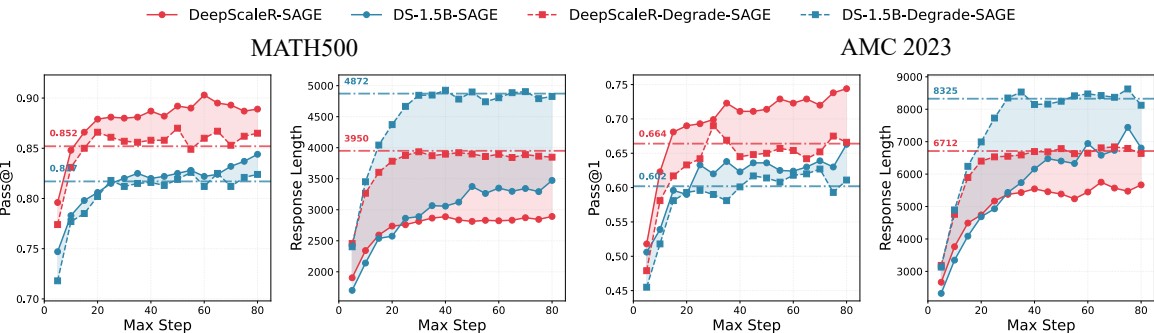

*Figure 8.* Performance comparison with SAGE and Degrade-SAGE on MATH-500 and AMC23 under different generation step budgets.

In summary, when provided with adequate exploration space, LRMs can identify precise and concise reasoning chains with high confidence and appropriately terminate the reasoning process, indicating that these models possess an inherent sense of when to stop reasoning. By contrast, current purely sampling-based strategies implicitly limit this capability of LRMs by relying solely on the next-token probability distribution.

## 5. Self-Aware Guided Efficient Reasoning

### 5.1. Methodology

While Section 4.3 demonstrates that TSearch w/ $\Phi$ effectively unleashes the efficient reasoning potential of LRMs as the exploration space expands, the method remains inherently **greedy**. Our goal, however, is to translate this insight into **random sampling**–based inference paradigms. Fortunately, from prior analysis in Section 4.2, when $\Phi$ is present, </think> consistently achieves the top rank upon appearance. This observation implies that TSearch w/ $\Phi$ is effectively equivalent to directly identifying reasoning steps that terminate with </think>, rendering token-level reasoning chain expansion unnecessary. Based on this observation and built upon TSearch w/ $\Phi$, we introduce **S**elf-**A**ware **G**uided **E**fficient Reasoning (**SAGE**), a simple yet effective sampling paradigm that performs *step-wise* reasoning chain expansion, with a more detailed motivation of this token-to-step transition in Appendix C. SAGE differs from TSearch w/ $\Phi$ in only the following two respects:

**Step-Wise Reasoning Chain Exploration.** At step $i$, each candidate sequence is extended by one full reasoning step **r** until the maximum reasoning step limit $T_{\max}$ is reached:

$$\mathbf{y}_{\leq i}^{(j,k)} = \mathbf{y}_{\leq i-1}^{(j)} \oplus \mathbf{r}_i^{(j,k)}, \quad \mathbf{r}_i^{(j,k)} \in \mathcal{R}^{(j)}, \qquad (8)$$

where $\mathcal{R}^{(j)} \triangleq \{\mathbf{r}_i^{(j,1)}, \mathbf{r}_i^{(j,2)}, \dots, \mathbf{r}_i^{(j,2m)}\}$ denotes the set of $2m$ reasoning steps independently sampled from the policy $\pi_\theta$ conditioned on the query **x** and prefix $\mathbf{y}_{\leq i-1}^{(j)}$ using vanilla random sampling. This process replaces the token-level

expansion in Equation 5.

**Exploration Termination.** Based on the conclusions in Section 4.2, we no longer need to manually set the tolerance rank ratio **TR** as the high-confidence reasoning branches consistently lead to confident ends. Our termination condition can be simply defined as : If we have reached a candidate sequence $\mathbf{y}_{\leq i}^{(j,k)}$, where $\mathbf{r}_k^{(j,k)}$ ends with </think> , we add it as a completion to the candidate sequence set $\mathcal{O}$.

### 5.2. SAGE Inference Scaling Trends with Step Budget

We introduce a step-wise alternative to random sampling namely Degrade SAGE to ablate the exploration space of SAGE. Degrade SAGE directly samples one reasoning step at each iteration until </think> appears or $T_{\max}$ is reached. To balance computational efficiency and performance (discussed in Appendix E.4), we adopt SAGE (2,1) as the representative of our algorithm. We scale the maximum reasoning step budget gradually and compare the pass@1 and response length of SAGE and Degraded SAGE on MATH-500 (mean@4) and AMC23 (mean@16) respectively. We mark *Random* results for DeepScaleR and DS-1.5B at 32,768 token budget with red and blue dashed lines, respectively.

**(1) The inference scaling trends of SAGE demonstrate the model's capability to terminate thinking at appropriate timings.** Under constrained step budgets, SAGE outperforms Degraded SAGE in pass@1 with similar sequence lengths. This advantage stems from SAGE stopping thinking earlier, leading to more complete CoTs. When step budgets are ample, a relatively stable performance gap emerges between SAGE and Degraded SAGE. Here, with token count no longer a bottleneck, the difference stem solely from reasoning chain exploration. These results clearly show that *SAGE effectively identifies reasoning chains superior to those of Degraded SAGE, as they are both shorter and more likely to lead to correct answers.*

**(2) SAGE prioritizes performance for strong models and hard datasets, and efficiency for weaker models and simple datasets.** On stronger DeepScale and harder AMC23,

*Table 2.* Pass@1, response length (LEN) and token efficiency (TE) results on four complex mathematical benchmarks. TE is calculated as Pass@1 / LEN. **Bold** and underlined denote the best and second-best results.

| Method | MATH-500 | | | AIME 2024 | | | AIME 2025 | | | OlympiadBench | | |
|---|---|---|---|---|---|---|---|---|---|---|---|---|
| | Pass@1↑(%) | LEN↓ | TE↑($\times 10^{-3}$) | Pass@1↑(%) | LEN↓ | TE↑($\times 10^{-3}$) | Pass@1↑(%) | LEN↓ | TE↑($\times 10^{-3}$) | Pass@1↑(%) | LEN↓ | TE↑($\times 10^{-3}$) |
| **DS-1.5B** | 83.2 | 4882 | 17.0 | 25.1 | 12300 | 2.04 | 20.9 | 11669 | 1.79 | 33.4 | 8954 | 3.73 |
| + LC-R1 | 80.4 (↓2.8) | 2973 (↓1909) | 27.0 (↑58.8%) | 23.3 (↓1.8) | 7098 (↓5202) | 3.28 (↑60.8%) | 20.9 (↑0.0) | 6942 (↓4727) | 3.01 (↑68.2%) | 32.0 (↓1.4) | 4632 (↓4322) | 6.91 (↑85.3%) |
| + ThinkPrune-2k | 81.7 (↓1.5) | 2826 (↓2056) | 28.9 (↑70.0%) | 23.7 (↓1.4) | 7085 (↓5215) | 3.35 (↑64.2%) | 19.7 (↓1.2) | 6918 (↓4751) | 2.85 (↑59.2%) | 32.9 (↓0.5) | 4752 (↓4202) | 6.92 (↑85.5%) |
| + AdaptThink | 80.4 (↓2.8) | **2563** (↓2319) | **31.4** (↑84.1%) | 25.7 (↑0.6) | 8055 (↓4245) | 3.19 (↑56.4%) | 21.8 (↑0.9) | 8155 (↓3514) | 2.67 (↑49.2%) | 32.6 (↓0.8) | **4563** (↓4391) | 7.14 (↑91.4%) |
| + Efficient Reasoning | 82.0 (↓1.2) | 2821 (↓2061) | 29.1 (↑70.6%) | 26.2 (↑1.1) | 9189 (↓3111) | 2.85 (↑39.7%) | 22.9 (↑2.0) | 8590 (↓3079) | 2.67 (↑49.2%) | 33.8 (↑0.4) | 5755 (↓3199) | 5.87 (↑57.4%) |
| + GRPO | 83.6 (↑0.4) | 3907 (↓975) | 21.4 (↑25.6%) | 28.3 (↑3.2) | 8767 (↓3533) | 3.23 (↑58.3%) | 24.1 (↑3.2) | 8263 (↓3406) | 2.92 (↑63.1%) | 34.2 (↑0.8) | 6323 (↓2631) | 5.41 (↑45.0%) |
| **+ SAGE-GRPO** | 84.8 (↑1.6) | 2915 (↓1967) | 29.1 (↑70.7%) | **28.8** (↑3.7) | 7243 (↓5057) | 3.98 (↑95.1%) | 26.5 (↑5.6) | 7479 (↓4190) | 3.54 (↑97.8%) | 36.9 (↑3.5) | 5050 (↓3904) | **7.31** (↑96.0%) |
| + GSPO | 83.4 (↑0.2) | 3898 (↓984) | 25.3 (↑21.4%) | 28.3 (↑3.2) | 8604 (↓3696) | 3.29 (↑61.3%) | 25.1 (↑4.2) | 8227 (↓3442) | 3.05 (↑70.4%) | 34.6 (↑1.2) | 6410 (↓2544) | 5.40 (↑44.8%) |
| **+ SAGE-GSPO** | **85.2** (↑2.0) | 2921 (↓1961) | 29.2 (↑71.6%) | 28.5 (↑3.4) | **6889** (↓5411) | **4.14** (↑102.9%) | **27.1** (↑6.2) | 7167 (↓4502) | **3.78** (↑111.1%) | **37.3** (↑3.9) | 5172 (↓3782) | 7.21 (↑93.3%) |
| **DeepScaleR** | 86.0 | 3805 | 22.6 | 31.4 | 9370 | 3.35 | 25.4 | 9310 | 2.73 | 35.9 | 5972 | 6.01 |
| + ThinkPrune-2k | 82.5 (↓3.5) | **2946** (↓859) | 28.0 (↑23.9%) | 33.5 (↑2.1) | 8108 (↓1262) | 4.13 (↑23.3%) | 26.0 (↑0.6) | 7486 (↓1824) | 3.47 (↑27.1%) | 35.1 (↓0.8) | **4723** (↓1249) | 7.43 (↑23.6%) |
| + GRPO | 87.6 (↑1.6) | 3482 (↓323) | 25.2 (↑11.3%) | 35.6 (↑4.2) | 8592 (↓778) | 4.14 (↑23.6%) | 27.4 (↑2.0) | 8185 (↓1125) | 3.35 (↑22.7%) | 36.2 (↑0.3) | 5443 (↓529) | 6.65 (↑10.6%) |
| **+ SAGE-GRPO** | **88.8** (↑2.8) | 3117 (↓688) | 28.4 (↑25.7%) | 36.1 (↑4.7) | 8094 (↓1276) | 4.46 (↑33.1%) | 27.2 (↑1.8) | 7704 (↓1606) | 3.53 (↑29.3%) | 36.5 (↑0.6) | 4890 (↓1082) | 7.46 (↑24.1%) |
| **DS-7B** | 91.6 | 3871 | 23.7 | 51.9 | 11305 | 4.59 | 37.1 | 12540 | 2.96 | 39.8 | 7839 | 5.08 |
| + LC-R1 | 87.3 (↓4.3) | **2076** (↓1795) | 42.1 (↑77.7%) | 51.7 (↓0.2) | 6820 (↓4485) | 7.58 (↑65.1%) | 35.7 (↓1.4) | 7458 (↓5082) | 4.79 (↑61.8%) | 41.4 (↑1.6) | 4193 (↓3646) | **9.87** (↑94.3%) |
| + AdaptThink | 88.9 (↓2.7) | 2199 (↓1672) | 40.4 (↑70.9%) | 52.1 (↑0.2) | 6679 (↓4626) | **7.80** (↑69.9%) | 35.0 (↓2.1) | 7807 (↓4733) | 4.48 (↑51.3%) | 38.9 (↑0.9) | 4915 (↓2924) | 7.91 (↑55.7%) |
| + Efficient Reasoning | 89.8 (↓1.8) | 2408 (↓1463) | 37.3 (↑57.6%) | 51.9 (↑0.0) | **6667** (↓4638) | 7.78 (↑69.5%) | 36.2 (↓0.9) | 7501 (↓5039) | 4.82 (↑62.8%) | 40.1 (↑0.3) | 4599 (↓3240) | 8.72 (↑71.7%) |
| + GRPO-LEAD | 89.5 (↓2.1) | 2752 (↓1119) | 32.5 (↑37.1%) | 53.1 (↑1.2) | 7023 (↓4282) | 7.56 (↑64.7%) | 36.1 (↓1.0) | 7842 (↓4698) | 4.60 (↑55.4%) | 40.6 (↑0.8) | 4972 (↓2867) | 8.17 (↑60.8%) |
| + GRPO | 92.0 (↑0.4) | 3219 (↓652) | 28.5 (↑20.2%) | 52.5 (↑0.6) | 8424 (↓2881) | 6.23 (↑35.7%) | 38.4 (↑1.3) | 10123 (↓2417) | 3.79 (↑28.0%) | 41.2 (↑1.4) | 5498 (↓2341) | 7.50 (↑47.6%) |
| **+ SAGE-GRPO** | **93.0** (↑1.4) | 2141 (↓1730) | **43.4** (↑83.1%) | **55.3** (↑3.4) | 6422 (↓4883) | **8.61** (↑87.6%) | 38.0 (↑0.9) | **6583** (↓5957) | **5.77** (↑94.9%) | **41.8** (↑2.0) | 4435 (↓3404) | 9.42 (↑85.4%) |
| **Qwen3-8B** | 94.4 | 5640 | 16.7 | 73.2 | 15920 | 4.60 | 67.3 | 18342 | 3.67 | 46.6 | 11707 | 4.00 |
| + GRPO | 93.6 (↓0.8) | 4470 (↓1170) | 20.9 (↑25.1%) | 72.8 (↓0.4) | 10573 (↓5347) | 6.89 (↑49.8%) | 66.6 (↓0.7) | 13981 (↓4361) | 4.76 (↑29.7%) | 45.1 (↓1.5) | 7512 (↓4195) | 6.00 (↑50.0%) |
| **+ SAGE-GRPO** | **95.0** (↑0.6) | 3015 (↓2625) | 31.5 (↑88.2%) | 73.5 (↑0.3) | 8975 (↓6945) | 8.19 (↑78.0%) | 66.6 (↓0.7) | 10052 (↓8290) | 6.58 (↑79.3%) | 45.4 (↓1.2) | 5972 (↓5735) | 7.60 (↑90.0%) |
| + GSPO | 94.6 (↑0.2) | 4342 (↓1298) | 22.2 (↑32.9%) | 73.0 (↓0.2) | 10544 (↓5376) | 6.92 (↑50.4%) | 66.2 (↓1.1) | 14082 (↓4260) | 4.70 (↑30.2%) | 46.6 (↑0.0) | 7964 (↓3743) | 5.85 (↑46.2%) |
| **+ SAGE-GSPO** | 94.4 (↑0.0) | 2753 (↓2887) | **34.3** (↑105.3%) | 73.7 (↑0.5) | 8547 (↓7373) | 8.62 (↑87.4%) | 66.0 (↓1.3) | 9183 (↓9159) | 7.19 (↑95.9%) | 46.7 (↑0.1) | 5436 (↓6271) | 8.59 (↑114.7%) |

we observe greater pass@1 gains. In contrast, on weaker DS-1.5B and simpler MATH-500, we note larger response length reductions. **From the model's perspective**, stronger models have a higher capability ceiling, enabling SAGE to deliver larger accuracy gains with more necessary tokens. In contrast, weaker models suffer from more severe overthinking, creating more chances for token redundancy reduction. **From the dataset's perspective**, LRMs can solve most problems on easier datasets, making response length the key optimization goal. By exploiting the model's inherent sense of when to stop thinking, SAGE identifies shorter reasoning chains to reduce response length significantly. Conversely, harder datasets contain more challenging problems requiring more tokens to solve, and SAGE boosts accuracy notably on them, confirming its efficacy on uncovering correct reasoning chains with minimal necessary tokens.

## 6. SAGE-RL: Integrating Efficient Reasoning Patterns into Current Inference Paradigms

As shown in Section 5, SAGE effectively unleashes reasoning models' implicit capacity for efficient reasoning. An appealing extension is to incorporate the efficient reasoning pattern uncovered by SAGE into standard pass@1 inference. Thus, we introduce **SAGE-RL**, a simple modification to RLVR, to achieve this goal.

Given a question $q$, RLVR typically samples a group of responses $\mathcal{G} = \{o_1, \ldots, o_G\}$ from the current policy. The sole difference between SAGE-RL and RLVR lies in the rollout phase, where SAGE-RL employs a hybrid sampling strategy. SAGE-RL employs **S**AGE (m,r) to generate $r$

responses $\{o_1^S, o_2^S, \ldots, o_r^S\}$ and uses standard **r**andom sampling for the remaining $G-r$ responses $\{o_1^R, o_2^R, \ldots, o_{G-r}^R\}$. Ultimately, the rollout phase in SAGE-RL yields the set of responses $\mathcal{G} = \{o_1^S, \ldots, o_r^S, o_1^R, \ldots, o_{G-r}^R\}$ for each $q$.

## 7. Experiments

We apply both RLVR (GRPO (Shao et al., 2024), GSPO (Zheng et al., 2025a) ) and corresponding SAGE-RL method (SAGE-GRPO, SAGE-GSPO) to tune four widely adopted LRMs with a group size of $G = 8$. The training objectives of these algorithms can be found in Appendix D.1. Within each group, SAGE-RL employs SAGE (2,2) to search for two completions with precise reasoning chains, while the remaining six completions are obtained through default random sampling in verl. We also compare with existing open-source methods, including LC-R1 (Cheng et al., 2025), ThinkPrune (Hou et al., 2025), AdaptThink (Zhang et al., 2025), Efficient-Reasoning (Arora et al., 2025), and GRPO-LEAD (Zhang et al., 2025). Additional implementation details are provided in Appendix D.2 due to space constraints.

### 7.1. Main Results

Table 2 presents a performance comparison among SAGE-RL and baselines. Due to space constraints, we present results from only four out of the six evaluated datasets. The complete experimental results and additional analysis are provided in Appendix E.

**(1) SAGE-RL achieves comprehensive improvements in both reasoning capability and token efficiency.** As

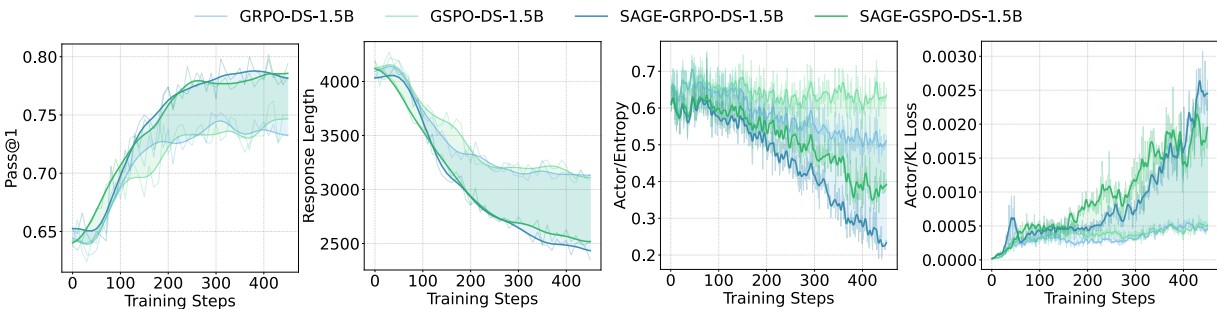

*Figure 9.* Training Dynamics comparison between RLVR and SAGE-RL. The left two figures present results evaluated every 10 steps on MATH-500 under an 8,192 token budget. The right two figures illustrate the entropy and KL divergence of the policy for every step.

shown in Table 2, most baselines achieve token compression at the cost of reduced reasoning capability. For instance, on MATH-500, AdaptThink compresses the token count of DS-1.5B from 4,882 to 2,563, but at the expense of a 2.8% drop in pass@1. Similar performance degradation is also widely observed across AIME 2024, AIME 2025 and OlympiaBench. RLVR was initially proposed to improve reasoning performance through extended reasoning lengths (DeepSeek-AI, 2025), yet existing baselines compromise this capability to different extents.

In contrast, SAGE-RL consistently achieve the best or second-best token efficiency across all benchmarks, while effectively improving the base models' capabilities on these complex reasoning tasks. This is because SAGE-RL achieves efficient reasoning by enabling LRMs to learn more precise reasoning chains, simultaneously shortening the inference trajectories while enhancing reasoning capability. As illustrated in Figure 8, the reasoning chains sampled by SAGE are shorter than those from standard sampling and more effectively guide the model toward correct solutions. In group-based comparison processes similar to GRPO, this advantage is amplified by the baseline's regularization. Since SAGE more frequently yields high-reward outcomes, the policy model naturally shifts its reasoning patterns toward the efficient modes discovered by SAGE.

**(2) SAGE-RL effectively enables LRMs to learn efficient reasoning patterns.** As shown in Table 2, although vanilla GRPO and GSPO moderately improve the reasoning capability of LRMs compared to other baselines, the inference trajectories learned by LRMs from standard random sampling still contain substantial token redundancy. Consequently, the overall token efficiency remains significantly lower than that of efficient reasoning baselines. In contrast, SAGE-RL achieves substantial improvements in both reasoning capability and token efficiency. Since the only difference lies in the sampling strategy for 2 out of 8 samples per group, the results demonstrate that SAGE-RL effectively enables the policy model to learn shorter yet more accurate reasoning patterns.

Figure 9 clearly illustrates this process. As training progresses, deploying SAGE-RL on both GRPO and GSPO leads to more pronounced improvements in pass@1 and greater reductions in response length. In contrast to standard RLVR, SAGE-RL shows a more significant entropy reduction, suggesting that the policy model gradually acquires the precise reasoning chains identified by SAGE, resulting in greater confidence during inference as training progresses. In terms of KL divergence, SAGE-RL also exhibits a more pronounced increasing trend. This indicates that the policy model deviates more significantly from the original probability distribution as training progresses. Such behavior suggests that the reasoning chains generated by SAGE, compared to those from random sampling, induce larger updates in the model. This is primarily because unleashing the model's efficient reasoning capability requires more substantial updates to learn reasoning patterns that differ markedly from the original ones.

As SAGE-RL's improvement solely stems from the rollout phase, the direct comparison with RLVR in this section serves as an effective **ablation study** of our approach.

### 7.2. Computational Cost Analysis

The additional cost of SAGE-RL stems from replacing 2 of 8 standard rollouts with SAGE(2,2). SAGE(2,2) keeps 4 candidate sequences, introducing 4× GPU memory overhead for these two trajectories. With sufficient parallelism, the wall-clock time is comparable. However, SAGE's step-by-step inference requires repeated calls to vLLM's generate operator, reducing the efficiency that standard random sampling enjoys. Moreover, as shown in Figure 9, SAGE-RL drives faster convergence toward shorter outputs, which significantly reduces rollout latency.

We report the overhead results for DS-1.5B. To ablate the impact of vLLM, we also implemented SAGE-GRPO using native HF inference. we compare the training cost of SAGE-GRPO with several baselines. As shown in Table 3, SAGE-GRPO introduces only moderate additional wall-clock cost under verl, while maintaining nearly identical

*Table 3.* Training Overhead Comparison.

| Method | Peak Memory (MiB) | Time (h) |
|---|---|---|
| GRPO | 70089 | 27.1 |
| SAGE-GRPO | 70091 | 30.2 |
| ThinkPrune | 69991 | 29.5 |
| LC-R1 | 70403 | 31.8 |
| AdaptThink | 75613 | 33.8 |

peak memory usage to GRPO. Overall, these results show that SAGE-RL not only improves the main mathematical reasoning benchmarks, but also generalizes to broader reasoning and coding scenarios with competitive training overhead. This supports the central claim that SAGE discovers efficient reasoning patterns that can be internalized by the policy and retained under standard inference.

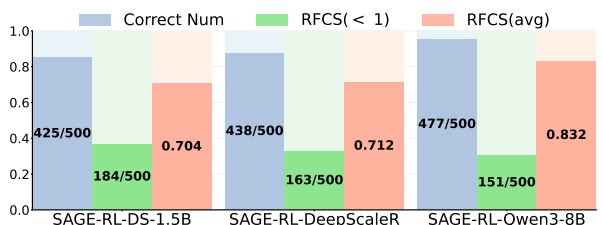

*Figure 10.* Statistics of RFCS on MATH-500 across different SAGE-RL-tuned models.

### 7.3. Analysis on Reasoning Behavior

We computed the RFCS metric on MATH-500 for SAGE-GRPO-tuned models, with results shown in Figure 10. Across all models, the proportion of samples with RFCS(<1) decreases substantially compared to Figure 3, indicating a significant reduction in redundant reasoning steps. Simultaneously, the RFCS(avg) increases markedly, suggesting that the reasoning models more frequently terminate thinking immediately after producing the correct answer. As shown in Figure 19 and Figure 20, SAGE-GRPO-tuned models effectively avoid generating a large number of ineffective reasoning steps. These findings strongly confirm that SAGE-RL effectively teaches LRMs precise reasoning patterns.

## 8. Conclusion

In this work, we uncover and demonstrate that LRMs implicitly know the appropriate time to stop thinking, but this potential is obscured by current sampling paradigms. Built on this observation, We propose SAGE, a sampling paradigm that unleash this capability to uncover precise reasoning chains, yielding significantly CoT length reduction and accuracy improvement. By simply integrating SAGE into the rollout process of RLVR, SAGE-RL achieves lasting gains in inference-time reasoning efficiency.

## Impact Statement

This paper uncovers and demonstrates the inherent efficient reasoning potential of LRMs, contributing to the broader field of Machine Learning. There are many potential societal consequences of our work, none of which we feel must be specifically highlighted here.

## Acknowledgements

This research was supported by NSFC (No. 62276015, No. 62506024) and GW2025-09.

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

# A. Related Work

## A.1. Stimulating Reasoning Capabilities through Reinforcement Learning

The introduction of OpenAI o1 (OpenAI, 2025) marks a major advance in reasoning performance and the beginning of the LRM era, inspiring efforts to replicate such strong reasoning abilities(Chen et al., 2026b; Zou et al., 2025; Chen et al., 2025d; He et al., 2025). DeepSeek-R1, for example, achieves comparable results using a simple rule-based reward with the group relative policy optimization (GRPO)(Shao et al., 2024) algorithm, and its open-source release has established RLVR (DeepSeek-AI, 2025; Kimi Team, 2025b; Gao et al., 2024; Lambert et al., 2025; Zeng et al., 2025; Wen et al., 2025a; Song et al., 2025) as an effective paradigm for improving LLM reasoning, and Yang et al. (2026) provides a principled theoretical analysis of its advantage estimation. This paradigm simplifies reward design by employing binary 0/1 rewards determined through rule-based correctness evaluation, eliminating the need for separate reward models as required in original GRPO (Shao et al., 2024; DeepSeek-AI, 2025; Schulman et al., 2017; Huang et al., 2026) implementations, thereby substantially reducing memory and computational overhead during RL training.

Subsequent models, including the Kimi K series (Kimi Team, 2025b;a), QwQ (Qwen Team, 2025), and O3 (OpenAI, 2025b), further advance these capabilities. RLVR assigns scores to trajectories based on pre-designed rules, rewarding desirable behaviors and penalizing undesirable ones. This encourages models to generate long CoTs to maximize correctness, fostering advanced reasoning behaviors such as search and backtracking. However, this also engenders a bias toward redundancy over the risk of error, which results in overthinking—wasting computational resources, impairing model performance, and ultimately limiting the practical applicability of LRMs.

In addition to the methods discussed above, a wide range of advanced techniques have been proposed in recent years to address various challenges in representation learning, model optimization, reasoning, and generative modeling. These include progress in interpretable representation learning (Li et al., 2025a), prompt-based structural modeling (Li et al., 2025c), diffusion-driven restoration (Li et al., 2025b), efficient transformer architectures for visual modeling (Fu et al., 2022), prompt-guided sequence modeling (Cai et al., 2023; 2024), parameter-efficient tuning strategies (Cai et al., 2025a), and novel normalization mechanisms for improving model stability (Cai et al., 2025b). Recent studies have also explored prototype-based medical diagnosis and medical vision-language reasoning (Zhu et al., 2025; 2026a; Lin et al., 2026; Zhu et al., 2026b), fairness-aware recommendation and graph domain adaptation (Chen et al., 2024b; 2025b; 2026a; Yuan et al., 2025), as well as efficient reasoning and reward-guided policy optimization for large language models (Yang et al., 2025c; Wang et al., 2026; Ding et al., 2026; Fang et al., 2026b;a). Although these works are designed for different task scenarios, they collectively enrich the toolkit of modern machine learning research and provide useful insights for understanding the generalization and optimization of neural models.

## A.2. Explorations in Efficient Reasoning

Overthinking issue is first identified and analyzed by Chen et al. (2025c), who observe that LRMs generate lengthy outputs that neither improve accuracy nor introduce new solution strategies especially for easy prompt. To address this, various works explore efficient reasoning from different angles.

**Training-Free Methods** typically improve reasoning efficiency through prompting engineering (Han et al., 2024; Xu et al., 2025a; Lee et al., 2025; Renze & Guven, 2024; Chen et al., 2024a; Aytes et al., 2025; Chuang et al., 2024; Ong et al., 2024; Xu et al., 2025b; Huang et al., 2025b; Han et al., 2025), Best-of-N sampling pruning (Xie et al., 2023; Liao et al., 2025) and optimizations (Li et al., 2024; Manvi et al., 2024; Aggarwal et al., 2023) , and early-exit (Ma et al., 2025a; Yang et al., 2025b; Fan et al., 2025) mechanisms during reasoning. These approaches cannot fundamentally resolve the issue of redundant reasoning in models, and their effectiveness is often heavily contingent upon the model's instruction-following capability. In practice, the observed improvements in experiments are typically modest or insignificant.

> While SAGE itself is also a training-free algorithm, it essentially serves to unleash the model's inherent potential for efficient reasoning. This allows the LRMs to select the currently optimal candidate sequence based on its self-aware at each inference iteration step.

**Offline Training Methods** primarily supervised fine-tuning models with variable-length CoT data (Yu et al., 2024; Kang et al., 2025; Xia et al., 2025; Ma et al., 2025b; Munkhbat et al., 2025; Liu et al., 2024; Han et al., 2024). Recently, ConCISE (Qiao et al., 2025) constructs concise CoT data by inserting prompt tokens and employing early-exit during inference, then enhances the model's reasoning conciseness through SFT/SimPO (Rafailov et al., 2024; Meng et al., 2024). The primary

challenge of this line of work lies in the difficulty of obtaining high-quality short chains of thought, and the offline training paradigm tends to limit the model's exploration ability on difficult problems.

> For similar reasons, we do not choose offline distillation to learn trajectories sampled by SAGE in this work. Since distillation depends on a strong teacher model, we are concerned that self-distillation will limit the upper boundary of the model's reasoning capability.

**Online Training Methods** mainly adopt reinforcement learning for better generalization. (Kimi Team, 2025b; Shen et al., 2025b; Cheng et al., 2025; Team et al., 2025; Luo et al., 2025a; Aggarwal & Welleck, 2025; Arora & Zanette, 2025; Yeo et al., 2025; Shen et al., 2025a; Qu et al., 2025; Cui et al., 2025) introduce length penalties in the reward function to suppress overly long reasoning traces. Yi et al. (2025), Hou et al. (2025), and Qi et al. (2025) optimize performance under a fixed token budget to balance efficiency and effectiveness. GFPO (Shrivastava et al., 2025) attains sampling outputs aligned with the optimization objective via oversampling. S-GRPO (Dai et al., 2025) and VSRM (Yue et al., 2025) truncate reasoning steps and perform repeated rollouts to evaluate the rewards of reasoning subchains, which are then leveraged for RL training. Zhang et al. (2025), Huang et al. (2025a), and Wu et al. (2025) assign predefined thinking patterns based on task difficulty, which essentially reflects a length budget. All the aforementioned methods are heavily rely on sophisticated reward design, which can easily lead to training instability or even reward hacking during the RL training process. Moreover, explicit or implicit integration of length compression into the optimization objective may impair the model's reasoning capabilities.

> In this work, instead of modifying the optimization objective, we optimize the sampling process to enable the policy model to directly learn the efficient reasoning chains uncovered by SAGE via the advantage estimation of RLVR. This design yields the following two key advantages: **(1) Low Computational Cost**: We eliminate the need for extra oversampling as in GFPO (Shrivastava et al., 2025), where a single parallel sampling step suffices to generate high-quality reasoning chains. Additionally, we do not require repeated rollouts for reward value estimation, a step essential to methods such as S-GRPO (Dai et al., 2025) and VSRM (Yue et al., 2025). **(2) Stable Training Dynamics**: By preserving all components of RLVR except for the rollout procedure, SAGE-RL exhibits no significant difference in training stability compared with vanilla RLVR.

## B. Significant Differences from Beam Search

In this section, we highlight the significant distinctions between TSearch w/ $\Phi$ and Beam Search from two perspectives: experimental results and underlying principles.

*Table 4.* Performance Comparison of different sampling strategies on different models (Max Tokens=10,086). Due to the inherent characteristic of Beam Search that it returns multiple responses by default, when calculating the ACC of Beam Search and TSearch, we consider a result correct if it contains at least one correct answer.

| Model | Sampling Strategy | ACC | LEN |
|---|---|---|---|
| DS-1.5B | Greedy | 0.81 | 4216 |
| | Random | 0.81 | 4142 |
| | Beam Search (4, 4) | 0.82 | 4472 |
| | TSearch w/ $\Phi$ (4, 4) | **0.84** | **2972** |
| Qwen3-8B | Greedy | 0.82 | 4505 |
| | Random | 0.82 | 4526 |
| | Beam Search (4, 4) | 0.84 | 4655 |
| | TSearch w/ $\Phi$ (4, 4) | **0.89** | **2946** |

We compared the performance of vanilla beam search with TSearch w/ $\Phi$ on a randomly selected subset of MATH-500 (size=100). For the fairness of comparison, we uniformly set the exploration width to $m = 4$. Since Beam Search directly returns the final set of candidate sequences, i.e., the number of returned sequences $r = m$, we therefore uniformly set $r = 4$. As shown in Table 4, Even though Beam Search generates four responses for each question, its final ACC is only comparable to those of random sampling and greedy sampling. Conversely, our algorithm achieves markedly higher accuracy while significantly reducing average response length.

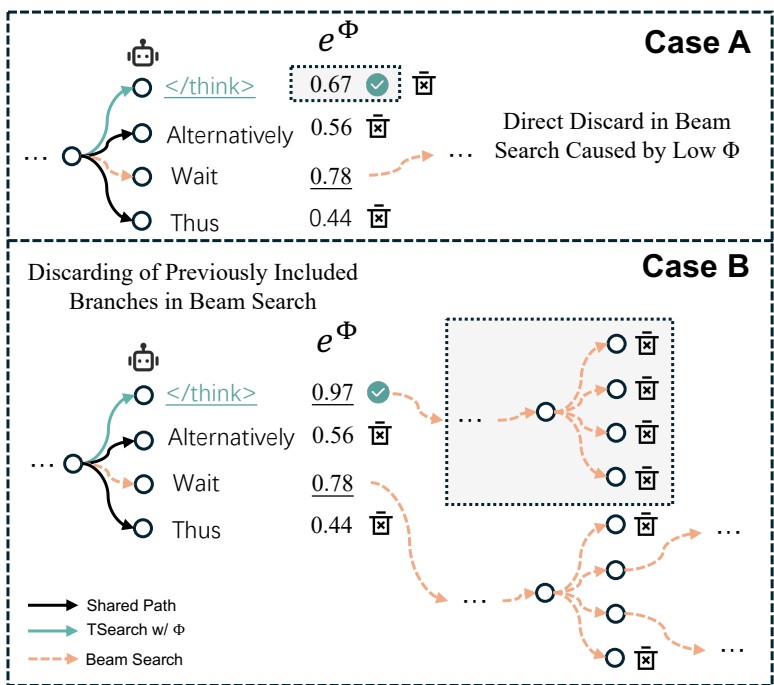

*Figure 11.* Two distinctions between TSearch w/ $\Phi$ and vanilla beam search.

We analyze and illustrate the root causes of these differences in Figure 11. **In Case A**, although </think> appears within the log-probability window, the corresponding candidate sequence is discarded because its overall confidence score $\Phi$ does not rank first. **In Case B**, a candidate sequence containing </think> is initially retained but is subsequently pruned during further expansion. In contrast, our algorithm directly accepts the sequence upon detecting </think> . These results indicate that our algorithm prevents the premature discarding of precise reasoning branches in later steps and significantly enhancing reasoning efficiency.

## C. The Motivation of Transition from Token-wise to Step-wise

Intuitively, steps serve as the logical reasoning units in LRMs, thus token-wise and step-wise search behave highly consistently. We further verify this via the following analyses.

**Termination Behavior:** We clarify that a reasoning step is defined as a complete reasoning segment generated by the model, ending at a natural step boundary or the end-of-thinking token. Since the end-of-thinking token appears at the end of a reasoning step rather than at an arbitrary token position, token-wise and step-wise exploration are aligned for the termination decision.

**Intermediate behavior:** We also examined whether token-level exploration inside a reasoning step is necessary. For each problem, we sampled complete-step prefixes and random-position prefixes, then ran TSearch from these prefixes and encoded candidates with all-MiniLM-L6-v2. Candidate branches starting from random positions inside a step were substantially more similar than those starting from complete steps, indicating that token-level search inside a step contributes little diversity.

**Overall performance:** Finally, we compared token-wise TSearch and step-wise SAGE under aligned settings. As Table 6 shows, SAGE closely matches TSearch in accuracy while avoiding expensive token-level candidate maintenance.

In summary, step-wise search behaves consistently with token-wise search, while lowering overhead by reducing candidate sequence maintenance. Thus, switching from TSearch to SAGE is natural and reasonable.

*Table 5.* Similarity among branches generated from different prefix types. Higher similarity for random-position prefixes suggests that token-level search inside reasoning steps is less useful than step-level branching.

| Prefix Type | Similarity |
|---|---|
| Complete steps | 0.6883 |
| Random positions | 0.8897 |

*Table 6.* Token-wise TSearch and step-wise SAGE under aligned settings.

| Method | TR | ACC | T-LEN | LEN |
|---|---|---|---|---|
| TSearch (4,1) w/ $\Phi$ | 1.00 | 0.92 | 2213 | 2609 |
| SAGE (4,1) | – | 0.92 | 2234 | 2655 |

# D. Experimental Details

## D.1. Objectives and Training Hyperparameters

The objectives of GRPO and SAGE-GRPO are as follows:

$$\mathcal{J}_{\text{GRPO}}(\theta) = \mathbb{E}_{x \sim \mathcal{D}, \{y_i\}_{i=1}^G \sim \pi_{\theta_{\text{old}}}(\cdot|x)} \left[ \frac{1}{G} \sum_{i=1}^{G} \frac{1}{|y_i|} \sum_{t=1}^{|y_i|} \min\left( w_{i,t}(\theta) \widehat{A}_{i,t}, \text{clip}\left( w_{i,t}(\theta), 1-\varepsilon, 1+\varepsilon \right) \widehat{A}_{i,t} \right) \right], \quad (9)$$

$$\mathcal{J}_{\text{SAGE-GRPO}}(\theta) = \mathbb{E}_{x \sim \mathcal{D}, \{y_i\}_{i=1}^G \sim \pi_{\theta_{\text{old}}}(\cdot|x)} \left[ \frac{1}{G} \left( \underbrace{\sum_{i=1}^{r} \frac{1}{|y_i|} \sum_{t=1}^{|y_i|} \min\left( w_{i,t}(\theta) \widehat{A}_{i,t}, \text{clip}\left( w_{i,t}(\theta), 1-\varepsilon, 1+\varepsilon \right) \widehat{A}_{i,t} \right)}_{\text{SAGE (m, r)}} + \right. \right.$$
$$\left. \left. \underbrace{\sum_{i=r+1}^{G} \frac{1}{|y_i|} \sum_{t=1}^{|y_i|} \min\left( w_{i,t}(\theta) \widehat{A}_{i,t}, \text{clip}\left( w_{i,t}(\theta), 1-\varepsilon, 1+\varepsilon \right) \widehat{A}_{i,t} \right)}_{\text{Random Sampling}} \right) \right] \quad (10)$$

where $G$ is the number of generated responses to each query $x$ (i.e., the group size), and the importance ratio $w_{i,t}(\theta)$ and advantage $\widehat{A}_{i,t}$ of token $y_{i,t}$ are:

$$w_{i,t}(\theta) = \frac{\pi_\theta(y_{i,t}|x, y_{i,<t})}{\pi_{\theta_{\text{old}}}(y_{i,t}|x, y_{i,<t})}, \quad \widehat{A}_{i,t} = \widehat{A}_i = \frac{r(x, y_i) - \text{mean}\left( \{r(x, y_i)\}_{i=1}^G \right)}{\text{std}\left( \{r(x, y_i)\}_{i=1}^G \right)}, \quad (11)$$

The objectives of GSPO and SAGE-GSPO are as follows:

$$\mathcal{J}_{\text{GSPO}}(\theta) = \mathbb{E}_{x \sim \mathcal{D}, \{y_i\}_{i=1}^G \sim \pi_{\theta_{\text{old}}}(\cdot|x)} \left[ \frac{1}{G} \sum_{i=1}^{G} \min\left( s_i(\theta) \widehat{A}_i, \text{clip}\left( s_i(\theta), 1-\varepsilon, 1+\varepsilon \right) \widehat{A}_i \right) \right], \quad (12)$$

$$\mathcal{J}_{\text{SAGE-GSPO}}(\theta) = \mathbb{E}_{x \sim \mathcal{D}, \{y_i\}_{i=1}^G \sim \pi_{\theta_{\text{old}}}(\cdot|x)} \left[ \frac{1}{G} \left( \underbrace{\sum_{i=1}^{r} \min\left( s_i(\theta) \widehat{A}_i, \text{clip}\left( s_i(\theta), 1-\varepsilon, 1+\varepsilon \right) \widehat{A}_i \right)}_{\text{SAGE (m, r)}} + \right. \right.$$
$$\left. \left. \underbrace{\sum_{i=r+1}^{G} \min\left( s_i(\theta) \widehat{A}_i, \text{clip}\left( s_i(\theta), 1-\varepsilon, 1+\varepsilon \right) \widehat{A}_i \right)}_{\text{Random Sampling}} \right) \right] \quad (13)$$

where we adopt the group-based advantage estimation:

$$\widehat{A}_i = \frac{r(x, y_i) - \text{mean}\left(\{r(x, y_i)\}_{i=1}^G\right)}{\text{std}\left(\{r(x, y_i)\}_{i=1}^G\right)}, \tag{14}$$

and define the importance ratio $s_i(\theta)$ based on sequence likelihood:

$$s_i(\theta) = \left(\frac{\pi_\theta(y_i|x)}{\pi_{\theta_{\text{old}}}(y_i|x)}\right)^{\frac{1}{|y_i|}} = \exp\left(\frac{1}{|y_i|}\sum_{t=1}^{|y_i|} \log \frac{\pi_\theta(y_{i,t}|x, y_{i,<t})}{\pi_{\theta_{\text{old}}}(y_{i,t}|x, y_{i,<t})}\right). \tag{15}$$

### D.2. Experimental Setup

To thoroughly evaluate the effectiveness of SAGE-RL, we conduct experiments using several widely adopted LRMs as base models, including DeepSeek-R1-Distill-Qwen-1.5B (DS-1.5B), DeepSeek-R1-Distill-Qwen-7B (DS-7B) (DeepSeek-AI, 2025), DeepScaleR (Luo et al., 2025b), and Qwen3-8B (Yang et al., 2025a).

**Training Data** Considering the importance of training data quality (Huang et al., 2025c), we use the English subset of DAPO (Yu et al., 2025) as well as MATH (Hendrycks et al., 2021) problems with difficulty from level 3 to level 5 (Zheng et al., 2025b). This collection consists of approximately 20,000 carefully curated problems covering a wide range of difficulty levels.

**Training Configuration** We use the `verl` (Sheng et al., 2024) framework for SAGE-RL training using the rule based reward function. To ensure a completely fair comparison that highlights the role of SAGE in the rollout phase, we adopt identical hyperparameter settings for the same base model across SAGE-RL and all its baselines and variants. We tune the base models with a global batch size of 32 across 8 GPUs for 600 steps with the Adam optimizer with learning rate of 1e-6, cosine warmup for the first 50 steps, and sampling temperature $T = 1.0$. We apply KL regularization with $\beta = 0.001$ and an entropy coefficient of $\gamma = 0.001$. Our models are trained with 9,216 maximum context length, with 1,024 tokens reserved for the prompt.

**Sampling Strategy** We tune all models with a group size of $G = 8$. Within each group, SAGE-RL employs SAGE (2,2) to search for two completions with precise reasoning chains, while the remaining six completions are obtained through the default random sampling in `verl`.

**Evaluation** We follow previous work (Yue et al., 2025; Liu et al., 2025; Dai et al., 2025) and select a comprehensive set of benchmarks, AIME24, AIME25(Art of Problem Solving, 2024),OlympiadBench(Chaoqun et al., 2024), MATH-500, Minerva(Lewkowycz et al., 2022), and AMC23(Mathematical Association of America, 2023), providing broader coverage than previous studies. During evaluation, we set the maximum generation length at 32768 tokens, consistent with Hou et al. (2025)'s work and DeepSeek-R1. The temperature and top-p are set to 1.0 and 0.95, respectively. For all benchmarks, we report the average pass@1, response length(LEN) and token efficiency(TE) over N runs. Specifically, for OlympiadBench, Minerva and MATH-500 where the benchmark sizes are relatively large, we set N to 8; for the other benchmarks, we set N to 32 to reduce randomness.

## E. Additional Experimental Results

### E.1. Comparison with Extended Datasets and Additional Analysis

In this section, we present the complete evaluation results on six mathematical datasets. We divide the benchmarks into two groups of equal size. The three datasets in the upper part of Table 7 are more challenging than those in the lower part.

**Performance on DS-1.5B and DS-7B** For experiments with DS-1.5B as the base model, SAGE-RL consistently achieve the best or second-best performance across all benchmarks, while effectively improving the original model's capabilities across all six mathematical reasoning benchmarks. Notably, SAGE-GSPO yields significant pass@1 gains of 6.2 % on AIME 2025 and 8.2 % on AMC23. AdaptThink stands out as a powerful baseline, attaining the highest token efficiency on MATH-500 and Minerva, while exhibiting the most pronounced reasoning simplification on OlympiadBench, MATH-500, and Minerva, but this high level of conciseness restrict the model's ability to explore different solution strategies. As a result, AdaptThink struggle in terms of overall performance and consistently lag behind our method. As for the other baselines, both their performance and efficiency are generally less competitive compared to SAGE-RL.

*Table 7.* Pass@1, response length(LEN) and TE results on six benchmarks and four base models before and after LC-R1, ThinkPrune-2k, AdaptThink, Efficient Reasoning, GRPO-LEAD, GRPO, GSPO, SAGE-GRPO and SAGE-GSPO. TE is calculated as Pass@1/LEN. Bold and underlined numbers denote the best and second-best results. The percentage in parentheses after TE indicates the improvement compared with the base model.

| Method | AIME 2024 | | | AIME 2025 | | | OlympiadBench | | |
|---|---|---|---|---|---|---|---|---|---|
| | Pass@1↑(%) | LEN↓ | TE↑($\times10^{-3}$) | Pass@1↑(%) | LEN↓ | TE↑($\times10^{-3}$) | Pass@1↑(%) | LEN↓ | TE↑($\times10^{-3}$) |
| **DS-1.5B** | 25.1 | 12300 | 2.04 | 20.9 | 11669 | 1.79 | 33.4 | 8954 | 3.73 |
| + LC-R1 | 23.3 (↓1.8) | 7098 (↓5202) | 3.28 (↑60.8%) | 20.9 (↑0.0) | 6942 (↓4727) | 3.01 (↑68.2%) | 32.0 (↓1.4) | 4632 (↓4322) | 6.91 (↑85.3%) |
| + ThinkPrune-2k | 23.7 (↓1.4) | 7085 (↓5215) | 3.35 (↑64.2%) | 19.7 (↓1.2) | 6918 (↓4751) | 2.85 (↑59.2%) | 32.9 (↓0.5) | 4752 (↓4202) | 6.92 (↑85.5%) |
| + AdaptThink | 25.7 (↑0.6) | 8055 (↓4245) | 3.19 (↑56.4%) | 21.8 (↑0.9) | 8155 (↓3514) | 2.67 (↑49.2%) | 32.6 (↓0.8) | **4563** (↓4391) | 7.14 (↑91.4%) |
| + Efficient Reasoning | 26.2 (↑1.1) | 9189 (↓3111) | 2.85 (↑39.7%) | 22.9 (↑2.0) | 8590 (↓3079) | 2.67 (↑49.2%) | 33.8 (↑0.4) | 5755 (↓3199) | 5.87 (↑57.4%) |
| + GRPO | 28.3 (↑3.2) | 8767 (↓3533) | 3.23 (↑58.3%) | 24.1 (↑3.2) | 8263 (↓3406) | 2.92 (↑63.1%) | 34.2 (↑0.8) | 6323 (↓2631) | 5.41 (↑45.0%) |
| **+ SAGE-GRPO** | **28.8** (↑3.7) | 7243 (↓5057) | 3.98 (↑95.1%) | 26.5 (↑5.6) | 7479 (↓4190) | 3.54 (↑97.8%) | 36.9 (↑3.5) | 5050 (↓3904) | **7.31** (↑96.0%) |
| + GSPO | 28.3 (↑3.2) | 8604 (↓3696) | 3.29 (↑61.3%) | 25.1 (↑4.2) | 8227 (↓3442) | 3.05 (↑70.4%) | 34.6 (↑1.2) | 6410 (↓2544) | 5.40 (↑44.8%) |
| **+ SAGE-GSPO** | 28.5 (↑3.4) | **6889** (↓5411) | **4.14** (↑102.9%) | **27.1** (↑6.2) | **7167** (↓4502) | **3.78** (↑111.1%) | **37.3** (↑3.9) | 5172 (↓3782) | 7.21 (↑93.3%) |
| **DeepScaleR** | 31.4 | 9370 | 3.35 | 25.4 | 9310 | 2.73 | 35.9 | 5972 | 6.01 |
| + ThinkPrune-2k | 33.5 (↑2.1) | 8108 (↓1262) | 4.13 (↑23.3%) | 26.0 (↑0.6) | 7486 (↓1824) | 3.47 (↑27.1%) | 35.1 (↓0.8) | 4723 (↓1249) | 7.43 (↑23.6%) |
| + GRPO | 35.6 (↑4.2) | 8592 (↓778) | 4.14 (↑23.6%) | **27.4** (↑2.0) | 8185 (↓1125) | 3.35 (↑22.7%) | 36.2 (↑0.3) | 5443 (↓529) | 6.65 (↑10.6%) |
| **+ SAGE-GRPO** | **36.1** (↑4.7) | **8094** (↓1276) | **4.46** (↑33.1%) | 27.2 (↑1.8) | **7704** (↓1606) | **3.53** (↑29.3%) | **36.5** (↑0.6) | **4890** (↓1082) | **7.46** (↑24.1%) |
| **DS-7B** | 51.9 | 11305 | 4.59 | 37.1 | 12540 | 2.96 | 39.8 | 7839 | 5.08 |
| + LC-R1 | 51.7 (↓0.2) | 6820 (↓4485) | 7.58 (↑65.1%) | 35.7 (↓1.4) | 7458 (↓5082) | 4.79 (↑61.8%) | 41.4 (↑1.6) | 4193 (↓3646) | **9.87** (↑94.3%) |
| + AdaptThink | 52.1 (↑0.2) | 6679 (↓4626) | 7.80 (↑69.9%) | 35.0 (↓2.1) | 7807 (↓4733) | 4.48 (↑72.3%) | 38.9 (↓0.9) | 4915 (↓2924) | 7.91 (↑55.7%) |
| + Efficient Reasoning | 51.9 (↑0.0) | 6667 (↓4638) | 7.78 (↑69.5%) | 36.2 (↓0.9) | 7501 (↓5039) | 4.82 (↑62.8%) | 40.1 (↑0.3) | 4599 (↓3240) | 8.72 (↑71.7%) |
| + GRPO-LEAD | 53.1 (↑1.2) | 7023 (↓4282) | 7.56 (↑64.7%) | 36.1 (↓1.0) | 7842 (↓4698) | 4.60 (↑55.4%) | 40.6 (↑0.8) | 4972 (↓2867) | 8.17 (↑60.8%) |
| + GRPO | 52.5 (↑0.6) | 8424 (↓2881) | 6.23 (↑35.7%) | **38.4** (↑1.3) | 10123 (↓2417) | 3.79 (↑28.0%) | 41.2 (↑1.4) | 5498 (↓2341) | 7.50 (↑47.6%) |
| **+ SAGE-GRPO** | **55.3** (↑3.4) | **6422** (↓4883) | **8.61** (↑87.6%) | 38.0 (↓0.9) | **6583** (↓5957) | **5.77** (↑94.9%) | **41.8** (↑2.0) | 4435 (↓3404) | 9.42 (↑85.4%) |
| **Qwen3-8B** | 73.2 | 15920 | 4.60 | **67.3** | 18342 | 3.67 | 46.6 | 11707 | 4.00 |
| + GRPO | 72.8 (↓0.4) | 10573 (↓5347) | 6.89 (↑49.8%) | 66.6 (↓0.7) | 13981 (↓4361) | 4.76 (↑29.7%) | 45.1 (↓1.5) | 7512 (↓4195) | 6.00 (↑50.0%) |
| **+ SAGE-GRPO** | 73.5 (↓0.3) | 8975 (↓6945) | 8.19 (↑78.0%) | 66.6 (↓0.7) | 10052 (↓8290) | 6.58 (↑79.3%) | 45.4 (↓1.2) | 5972 (↓5735) | 7.60 (↑90.0%) |
| + GSPO | 73.0 (↓0.2) | 10544 (↓5376) | 6.92 (↑50.4%) | 66.2 (↓1.1) | 14082 (↓4260) | 4.70 (↑30.2%) | 46.6 (↑0.0) | 7964 (↓3743) | 5.85 (↑46.2%) |
| **+ SAGE-GSPO** | **73.7** (↑0.5) | **8547** (↓7373) | **8.62** (↑87.4%) | 66.0 (↓1.3) | **9183** (↓9159) | **7.19** (↑95.9%) | **46.7** (↑0.1) | **5436** (↓6271) | **8.59** (↑114.7%) |

| Method | MATH-500 | | | Minerva | | | AMC23 | | |
|---|---|---|---|---|---|---|---|---|---|
| | Pass@1↑(%) | LEN↓ | TE↑($\times10^{-3}$) | Pass@1↑(%) | LEN↓ | TE↑($\times10^{-3}$) | Pass@1↑(%) | LEN↓ | TE↑($\times10^{-3}$) |
| **DS-1.5B** | 83.2 | 4882 | 17.0 | 30.1 | 6210 | 4.85 | 60.1 | 8250 | 7.28 |
| + LC-R1 | 80.4 (↓2.8) | 2973 (↓1909) | 27.0 (↑58.8%) | 31.8 (↑1.7) | 3512 (↓2698) | 9.06 (↑86.8%) | 61.8 (↑1.7) | **4889** (↓3361) | 12.6 (↑73.6%) |
| + ThinkPrune-2k | 81.7 (↓1.5) | 2826 (↓2056) | 28.9 (↑70.0%) | 32.9 (↑2.8) | 3667 (↓2543) | 8.97 (↑85.0%) | 60.8 (↑0.7) | 5224 (↓3026) | 11.6 (↑59.9%) |
| + AdaptThink | 80.4 (↓2.8) | **2563** (↓2319) | **31.4** (↑84.1%) | 32.3 (↑2.2) | **2912** (↓3298) | **11.1** (↑128.7%) | 62.3 (↑2.2) | 4969 (↓3281) | 12.5 (↑71.7%) |
| + Efficient Reasoning | 82.0 (↓1.2) | 2821 (↓2061) | 29.1 (↑70.6%) | 31.4 (↑1.3) | 3530 (↓2680) | 8.90 (↑83.5%) | 64.7 (↑4.6) | 5202 (↓3048) | 12.4 (↑70.9%) |
| + GRPO | 83.6 (↑0.4) | 3907 (↓975) | 21.4 (↑25.6%) | 32.0 (↑1.9) | 4806 (↓1404) | 6.66 (↑37.3%) | 65.4 (↑5.3) | 5771 (↓2479) | 11.3 (↑55.6%) |
| **+ SAGE-GRPO** | 84.8 (↑1.6) | 2915 (↓1967) | 29.1 (↑70.7%) | **33.8** (↑3.7) | 3735 (↓2475) | 9.05 (↑86.6%) | 66.3 (↑6.2) | 5091 (↓3159) | **13.0** (↑78.9%) |
| + GSPO | 83.4 (↑0.2) | 3898 (↓984) | 25.3 (↑21.4%) | 32.0 (↑1.9) | 4454 (↓1756) | 7.18 (↑48.0%) | 66.1 (↑6.0) | 6095 (↓2191) | 10.9 (↑63.1%) |
| **+ SAGE-GSPO** | **85.2** (↑2.0) | 2921 (↓1961) | 29.2 (↑71.6%) | 33.6 (↑3.5) | 3647 (↓2563) | 9.21 (↑89.9%) | **68.3** (↑8.2) | 5278 (↓2972) | 12.9 (↑77.7%) |
| **DeepScaleR** | 86.0 | 3805 | 22.6 | 38.6 | 5184 | 7.45 | 64.2 | 6683 | 9.61 |
| + ThinkPrune-2k | 82.5 (↓3.5) | **2946** (↓859) | 28.0 (↑23.9%) | 37.9 (↓0.7) | **3188** (↓1996) | **11.9** (↑59.6%) | 65.8 (↑1.6) | **5046** (↓1637) | **13.0** (↑35.6%) |
| + GRPO | 87.6 (↑1.6) | 3482 (↓323) | 25.2 (↑11.3%) | 40.4 (↑1.8) | 4386 (↓798) | 9.21 (↑23.6%) | 69.3 (↑5.1) | 5872 (↓811) | 11.8 (↑22.8%) |
| **+ SAGE-GRPO** | **88.8** (↑2.8) | 3117 (↓688) | **28.4** (↑25.7%) | **41.4** (↑2.8) | 3817 (↓1367) | 10.9 (↑45.6%) | **70.9** (↑6.7) | 5438 (↓1245) | **13.0** (↑35.7%) |
| **DS-7B** | 91.6 | 3871 | 23.7 | 43.0 | 5490 | 7.83 | 81.9 | 7170 | 11.4 |
| + LC-R1 | 87.3 (↓4.3) | **2076** (↓1795) | 42.1 (↑77.7%) | 44.4 (↑1.4) | **2834** (↓2656) | 15.7 (↑100.0%) | 79.1 (↓2.8) | **3686** (↓3484) | 21.5 (↑87.9%) |
| + AdaptThink | 88.9 (↓2.7) | 2199 (↓1672) | 40.4 (↑70.9%) | 45.2 (↑2.2) | 2869 (↓2621) | 15.8 (↑101.2%) | 80.7 (↓1.2) | 5130 (↓2040) | 15.7 (↑37.7%) |
| + Efficient Reasoning | 89.8 (↓1.8) | 2408 (↓1463) | 37.3 (↑57.6%) | 45.7 (↑2.7) | 2903 (↓2587) | 15.7 (↑101.0%) | 80.7 (↓1.2) | 4933 (↓2237) | 16.4 (↑43.3%) |
| + GRPO-LEAD | 89.5 (↓2.1) | 2752 (↓1119) | 32.5 (↑37.1%) | **46.3** (↑3.3) | 2990 (↓2500) | 16.0 (↑104.3%) | 82.7 (↑0.8) | 4384 (↓2786) | 18.9 (↑65.8%) |
| + GRPO | 92.0 (↑0.4) | 3219 (↓652) | 28.5 (↑20.2%) | 46.0 (↑3.0) | 3510 (↓1980) | 13.1 (↑67.4%) | 83.0 (↑1.1) | 4880 (↓2290) | 17.0 (↑49.0%) |
| **+ SAGE-GRPO** | **93.0** (↑1.4) | 2141 (↓1730) | **43.4** (↑83.1%) | 45.2 (↑2.2) | 2692 (↓2798) | **16.8** (↑114.4%) | **84.9** (↑3.0) | 3953 (↓3217) | **21.5** (↑88.1%) |
| **Qwen3-8B** | 94.4 | 5640 | 16.7 | 51.8 | 7358 | 7.04 | 90.5 | 10852 | 8.34 |
| + GRPO | 93.6 (↓0.8) | 4470 (↓1170) | 20.9 (↑25.1%) | 52.6 (↑0.8) | 4964 (↓2394) | 10.6 (↑50.6%) | 88.6 (↓1.9) | 7079 (↓3773) | 12.5 (↑50.1%) |
| **+ SAGE-GRPO** | **95.0** (↑0.6) | 3015 (↓2625) | 31.5 (↑88.2%) | 53.5 (↑1.7) | 3390 (↓3968) | 15.8 (↑124.2%) | 90.7 (↑0.2) | 5563 (↓5289) | 16.3 (↑95.4%) |
| + GSPO | 94.6 (↑0.2) | 4342 (↓1298) | 22.2 (↑32.9%) | 49.6 (↓2.2) | 3962 (↓3396) | 12.5 (↑77.8%) | 87.7 (↓2.8) | 6464 (↓4388) | 13.6 (↑63.1%) |
| **+ SAGE-GSPO** | 94.4 (↑0.0) | **2753** (↓2887) | **34.3** (↑105.3%) | **53.7** (↑1.9) | **3363** (↓3995) | **16.0** (↑126.8%) | **90.9** (↑0.4) | **5041** (↓5811) | **18.0** (↑106.7%) |

A similar trend is observed when using DS-7B as the base model. Both GRPO-LEAD and Efficient-Reasoning adopt a strategy of sacrificing less compression in exchange for improved performance; however, the reasoning capability gains they achieve remain substantially smaller than those of SAGE-RL. For instance, on AIME 2024, SAGE-GRPO not only outperforms GRPO-LEAD by 2.2 % in pass@1, but also produces noticeably shorter responses.

These two sets of experiments together indicate that, on distilled models, SAGE-RL not only effectively alleviates the overthinking problem but also substantially enhances the model's reasoning capability on complex mathematical problems. Since our method simultaneously improves the base model's reasoning capability and the precision of its thinking process, it achieves more consistent and substantially larger gains in token efficiency compared to other approaches.

**Performance on DeepScaleR**  DeepScaleR has undergone systematic and comprehensive reinforcement learning (Luo et al., 2025b). Consequently, additional fine-tuning on this model typically yields only marginal performance gains, which accounts for the scarcity of related works that adopt DeepScaleR as a base model. Nevertheless, SAGE-RL still achieves relatively significant token efficiency improvements on DeepScaleR. Particularly on OlympiaBench, Math-500 and Minerva, SAGE-RL delivers roughly twice the token efficiency gains of GRPO, demonstrating that SAGE-RL yields relatively substantial benefits even for models with extensive post-training.

**Performance on Qwen3-8B**  As one of the strongest reasoning models under the same parameter scale, Qwen3-8B achieves excellent performance across various mathematical reasoning tasks. Even on the highly challenging AIME 2025, it attains an impressive pass@1 of 67.3%. However, as illustrated in Figure 3, the overthinking problem remains largely unresolved in this model. For instance, on MATH-500, despite comparable pass@1 performance, the average response length is more than 2.5 times that of SAGE-GRPO-tuned DS-7B.

Notably, vanilla RLVR is capable of moderately reducing the response length of the base model. This effect stems from the training procedure, where sequences must be padded to a fixed batch length, causing the token budget in evaluation to be significantly smaller than that used in inference. As a result, the model tends to receive positive rewards more readily for short answers, encouraging shorter generations. Nevertheless, this mechanism limits the model's ability to improve or may even cause declines on reasoning tasks of varying difficulty, particularly on datasets such as AIME 2024 and AMC 23.

In contrast, SAGE-RL still achieves moderate improvements in the reasoning capability of Qwen3-8B under limited training token budgets, while effectively reducing the redundancy in the thinking process. For example, SAGE-GSPO attains a 1.9% increase in pass@1 on Minerva, while compressing the average response length to only 45.7% of the original. These results strongly demonstrate that SAGE-RL remains highly effective even on state-of-the-art reasoning models.

**Comparison of SAGE-GRPO and SAGE-GSPO**  As shown in Figure 9, across both GRPO and GSPO, the key variations in pass@1, response length, and KL loss are driven by the incorporation of SAGE-RL rather than fundamental RLVR algorithms. This underscores the robust positive impact of our approach across different RLVR implementations. In terms of entropy, GSPO-tuned models show elevated values, largely due to sequence-level importance sampling disregarding fine-grained token-level variations, thereby resulting in higher inference uncertainty.

From the experimental results shown in Table 7, SAGE-GSPO exhibits particularly strong performance in reducing response length and slightly outperforms SAGE-GRPO in overall metrics. We hypothesize that this advantage stems from the greater stability of GSPO's sequence-level importance sampling compared to GRPO's token-level importance sampling, which is especially beneficial in more unstable scenarios such as MoE models (Zheng et al., 2025a).

A similar issue arises in SAGE-RL due to the hybrid sampling used in the rollout phase. In SAGE-GRPO, some of the rollouts are generated by selecting sequences at every reasoning step based on the full-sequence confidence score $\Phi$, rather than greedily choosing the highest log-probability token as in Equation 7. Consequently, as indicated in Equation 11, the probability $\pi_{\theta_{\text{old}}}(y_{i,t} \mid x, y_{i,<t})$ under the old policy may be lower than that of random sampling, increasing the likelihood of clipping during importance sampling. In contrast, GSPO treats the entire sequence as the basic unit for importance sampling, thereby avoiding this issue entirely.

Overall, Table 7 demonstrates that SAGE-RL achieves substantially superior performance compared to all baseline methods across six challenging mathematical reasoning tasks. Meanwhile, Figure 9 reveals higher pass@1 scores and increased KL divergence, accompanied by reduced response entropy and shorter response lengths. These results indicate that SAGE successfully unleashes the model's implicit capacity for timely thinking termination. Consequently, the model learns efficient reasoning with increased confidence, confirming the viability of leveraging RLVR to instill effective reasoning patterns. This is consistent with the results of Wen et al. (2025b), which demonstrate that RLVR effectively promotes correct reasoning chains in base LLMs.

### E.2. Hyperparameters Sensitivity Analysis

This section examines the influence of the two primary hyperparameters influencing SAGE-RL: the SAGE exploration width $m$ and the total number of rollouts $r$ produced by SAGE per group. We evaluate SAGE-RL under various combinations of these parameters and denote each setting as SAGE$(m, r)$-RL. Figure 12 illustrates the training dynamics of DS-1.5B with SAGE-GRPO under different hyperparameter combinations and GRPO. The corresponding evaluation results on four mathematical datasets are reported in Table 8.

*Table 8.* A comparison of experimental results for DS-1.5B under different SAGE-GRPO parameter settings. Here, SAGE (m, r) denotes an exploration width of $m$, with the final retention of $r$ different trajectories.

| Method | MATH-500 | | | AIME 2024 | | | AIME 2025 | | | OlympiadBench | | |
|---|---|---|---|---|---|---|---|---|---|---|---|---|
| | Pass@1↑(%) | LEN↓ | TE↑(×10⁻³) | Pass@1↑(%) | LEN↓ | TE↑(×10⁻³) | Pass@1↑(%) | LEN↓ | TE↑(×10⁻³) | Pass@1↑(%) | LEN↓ | TE↑(×10⁻³) |
| **DS-1.5B** | 83.2 | 4882 | 17.0 | 25.1 | 12300 | 2.04 | 20.9 | 11669 | 1.79 | 33.4 | 8954 | 3.73 |
| + GRPO | 83.6 (↑0.4) | 3907 (↓975) | 21.4 (↑25.6%) | 28.3 (↑3.2) | 8767 (↓3533) | 3.23 (↑58.3%) | 24.1 (↑3.2) | 8263 (↓3406) | 2.92 (↑63.1%) | 34.2 (↑0.8) | 6323 (↓2631) | 5.41 (↑45.0%) |
| + SAGE (1,1)-GRPO | 84.0 (↑0.8) | 3416 (↓1466) | 24.6 (↑44.7%) | 28.3 (↑3.2) | 7979 (↓4321) | 3.55 (↑74.0%) | 24.8 (↑3.9) | 7730 (↓3939) | 3.21 (↑79.3%) | 34.5 (↑1.1) | 5857 (↓3097) | 5.89 (↑57.9%) |
| + SAGE (2,1)-GRPO | 84.2 (↑1.0) | 2952 (↓1930) | 28.5 (↑67.8%) | 28.5 (↑3.4) | 7308 (↓4992) | 3.90 (↑91.2%) | 25.7 (↑4.8) | 7603 (↓4066) | 3.38 (↑88.8%) | 35.2 (↑1.8) | 5267 (↓3687) | 6.68 (↑79.1%) |
| + SAGE (2,2)-GRPO | **84.8** (↑1.6) | **2915** (↓1967) | **29.1** (↑70.7%) | **28.8** (↑3.7) | **7243** (↓5057) | **3.98** (↑95.1%) | **26.5** (↑5.6) | **7479** (↓4190) | **3.54** (↑97.8%) | **36.9** (↑3.5) | **5050** (↓3904) | **7.31** (↑96.0%) |

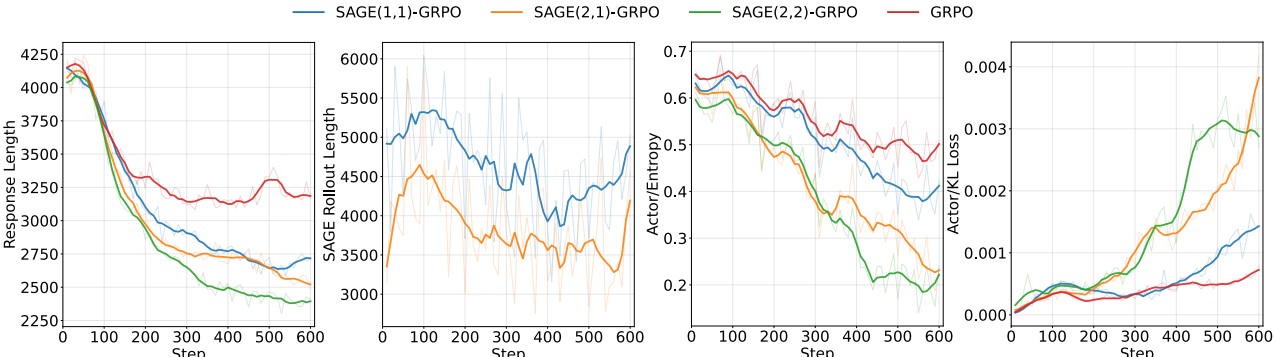

*Figure 12.* Training dynamics comparison for SAGE-GRPO with distinct hyperparameter combination: average response length when tested on MATH500, average SAGE-produced trajectory length during training, entropy, and KL divergence.

**The Impact of SAGE Rollout Quantity** As shown in Table 8, the transition of $r$ from 1 to 2 has limited effect on the results. From a policy optimization perspective, larger $r$ allows the policy model to learn from more efficient reasoning samples; however, the advantage estimate per sample becomes less sharp compared to $r = 1$, leading to similar overall updates.

Figure 12 shows that SAGE (2,1)-GRPO and SAGE (2,2)-GRPO display very similar trends in entropy and KL divergence, markedly different from those of SAGE (1,1)-GRPO and vanilla GRPO. This indicates that **enlarging $r$ has little impact on policy updates, as rollouts with similar reasoning trajectories offer minimal additional information**.

**The Impact of Exploration Width** On the other hand, enlarging $m$ from 1 to 2 yields substantial performance gains. According to the results shown in Table 8, while SAGE (1,1)-GRPO yields moderate improvements over the vanilla GRPO baseline, its performance is markedly inferior to that of SAGE (2,1)-GRPO.

This indicates that exploration width significantly influences the activation of the model's efficient reasoning capability, consistent with the findings in Figure 11. As illustrated in Figure 12, SAGE (1,1)-GRPO exhibits significantly milder entropy reduction and KL divergence increase relative to SAGE (2,1)-GRPO, and its training dynamics remain much closer to the vanilla GRPO. More directly, both the average length of SAGE-produced rollouts in SAGE(2,1)-GRPO and the average

response length of the model at test time are significantly shorter than those observed in SAGE(1,1)-GRPO. These results indicate that **a limited exploration width $m$ causes SAGE-RL to largely collapse to the standard GRPO optimization behavior**.

### E.3. SAGE-RL shows Promising Potential in Difficult Reasoning Tasks

To more clearly elucidate the operational mechanism behind SAGE-RL, we compare the training dynamics of SAGE-GRPO-DS-1.5B (Ours) and GRPO-DS-1.5B (GRPO) on MATH-500 across five difficulty levels as training steps scales. The level 1-5 ranges from low to high, reflecting increasing levels of difficulty.

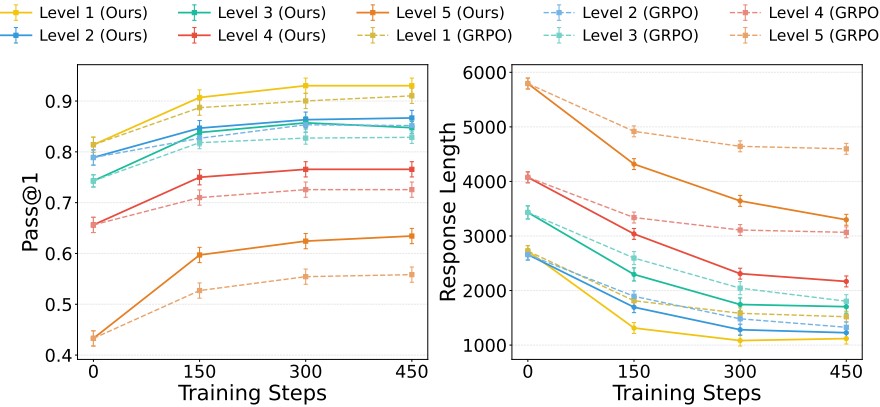

*Figure 13.* The training dynamics of SAGE-GRPO- DS-1.5B (Ours) and GRPO-DS-1.5B (GRPO) on MATH-500 across level 1-5.

As illustrated in Figure 13, both GRPO and SAGE-GRPO show steady performance gains across all difficulty levels of MATH500 as training progresses. SAGE-GRPO converges markedly faster than GRPO at every level and eventually attains performance comparable to GRPO on level 1-3 problems. **A clear divergence appears on level 4-5 problems, where SAGE-GRPO achieves substantially superior pass@1 and lower response length.** Remarkably, the downward trend in response length for SAGE-GRPO continues even after GRPO has converged.

These observations suggest that SAGE-RL primarily improves overall performance by dramatically increasing reasoning efficiency on difficult problems. This is consistent with the results in Table 7, which reveal significantly larger gains from SAGE-RL fine-tuning on more challenging benchmarks such as AIME 2024, AIME 2025, OlympiadBench, and Minerva than on relatively easier ones such as MATH-500 and AMC23.

Collectively, these findings highlight **the considerable potential of SAGE-RL in overcoming the reasoning performance bottlenecks faced by current LRMs on highly difficult tasks**.

### E.4. Time Complexity Analysis

**Time Complexity Analysis of SAGE**    SAGE generates $2m$ reasoning steps in parallel with a fixed exploration width $m$ at each expansion step. Therefore, it theoretically achieves the same time complexity as Degrade SAGE, meanwhile, its space complexity is approximately $2m$ times higher. However, as we adopt vLLM (Kwon et al., 2023) as the inference engine, whose core design philosophy centers on a space-for-time tradeoff: it maximizes GPU memory utilization to minimize inference latency. Nevertheless, our implementation is constrained to the use of only 8 GPUs. Under this memory-limited setting, SAGE incurs higher inference-time cost compared to Degrade SAGE.

We report the average per-sample runtime of SAGE (m, 1) under different EW in this constrained hardware environment. Here, $m$ denotes the exploration width EW. When $m$ = 0, SAGE degenerates to Degrade SAGE. As shown in Figure 14(a), the inference time of DS-1.5B remains consistently higher than that of DeepScaleR. Moreover, the average inference time per response increases significantly with larger exploration widths.

This primarily arises from the trade-off adopted by vLLM: elevated space complexity is exchanged for reduced time complexity in the context of limited computational resources. In particular, once the exploration width exceeds 2, the growth rate of inference time accelerates further. Therefore, **we primarily set exploration width $m$ = 2, which represents the transition**

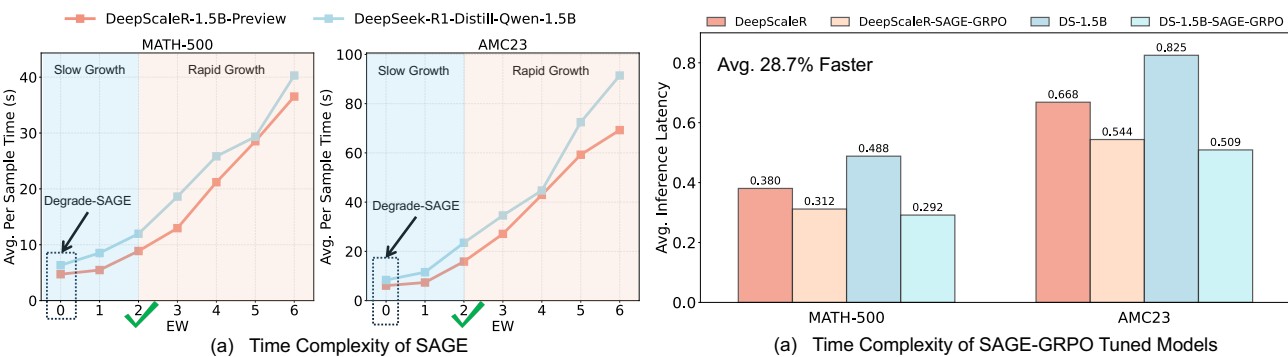

*Figure 14.* (a) Average inference time of SAGE on each question; (b) Comparison of Normalized inference time between the base models and the SAGE-GRPO tuned models on each question, approximated and normalized by the average response length.

**point between the slow-growth and fast-growth regions, to achieve a balanced trade-off between efficiency and performance.**

**Time Complexity Analysis of SAGE-RL Tuned Models**   In the standard pass@1 inference setting, the KV cache is prefilled during the initial prompt processing phase, which ensures that the generation latency per subsequent token remains approximately constant. Consequently, for short queries, the total inference time of each completion scales nearly linearly with the number of generated tokens. However, vLLM aggressively optimizes inference speed through techniques such as KV cache reuse and continuous batching, which compromises the fairness of direct wall-clock time comparisons.

Given the approximately linear relationship between inference time and the number of generated tokens, we adopt the proxy metric $0.0001 \times$ (average response length) to reflect the average inference latency. We compare this normalized metric between the base models and our SAGE-GRPO-tuned models. As shown in Figure 14(b), although SAGE incurs increasing inference-time cost with larger exploration widths under constrained hardware, SAGE-RL-tuned models can significantly reduce the average inference time in the standard pass@1 inference paradigm. Specifically, even on the relatively easier MATH-500 and AMC23 subsets among the six datasets we evaluated, our approach still achieves a 28.7% reduction in inference latency. When approximating average inference time using the average response length, Table 7 clearly shows that, compared to the baseline, **SAGE-RL-tuned models reduce inference latency by more than 40% across the majority of models and benchmarks.**

## F. Hyperparameter Ablation

We conducted hyperparameter ablations for SAGE-GRPO and additional baselines beyond GRPO, including AdaptThink and ThinkPrune-2k. We swept the learning rate over $\{5 \times 10^{-7}, 2 \times 10^{-6}\}$ and the KL coefficient over $\{5 \times 10^{-4}, 2 \times 10^{-3}\}$, training each method independently under each setting.

Across all four configurations, the relative ranking is stable: SAGE-GRPO consistently achieves the highest Pass@1 and token efficiency. AdaptThink and ThinkPrune obtain shorter responses in some cases, but this compression comes at an accuracy cost; both underperform vanilla GRPO in Pass@1. These results indicate that the improvements of SAGE-GRPO are not artifacts of a particular hyperparameter choice.

## G. Pseudo Code of SAGE

The example Python code of SAGE is illustrated in Figure 15, Figure 16 and Figure 17.

*Table 9.* Hyperparameter sweep over learning rate and KL coefficient on DS-1.5B. SAGE-GRPO consistently achieves the best Pass@1 and token efficiency on both AIME 2025 and OlympiadBench.

| LR / KL | Method | AIME 2025 | | | OlympiadBench | | |
|---|---|---|---|---|---|---|---|
| | | Pass@1 | LEN | TE ($\times 10^{-3}$) | Pass@1 | LEN | TE ($\times 10^{-3}$) |
| $5 \times 10^{-7}$ / $5 \times 10^{-4}$ | SAGE-GRPO | 25.8 | 7673 | 3.36 | 36.1 | 5253 | 6.87 |
| $5 \times 10^{-7}$ / $5 \times 10^{-4}$ | GRPO | 23.3 | 8461 | 2.75 | 33.4 | 6536 | 5.11 |
| $5 \times 10^{-7}$ / $5 \times 10^{-4}$ | AdaptThink | 21.9 | 8102 | 2.70 | 32.4 | 7306 | 6.85 |
| $5 \times 10^{-7}$ / $5 \times 10^{-4}$ | ThinkPrune | 19.7 | 6950 | 2.83 | 32.3 | 4793 | 6.74 |
| $5 \times 10^{-7}$ / $2 \times 10^{-3}$ | SAGE-GRPO | 25.5 | 7804 | 3.27 | 35.8 | 5387 | 6.65 |
| $5 \times 10^{-7}$ / $2 \times 10^{-3}$ | GRPO | 23.0 | 8598 | 2.68 | 33.1 | 6665 | 4.97 |
| $5 \times 10^{-7}$ / $2 \times 10^{-3}$ | AdaptThink | 22.0 | 8254 | 2.67 | 31.5 | 7626 | 6.61 |
| $5 \times 10^{-7}$ / $2 \times 10^{-3}$ | ThinkPrune | 19.9 | 6925 | 2.87 | 31.6 | 4779 | 6.61 |
| $2 \times 10^{-6}$ / $5 \times 10^{-4}$ | SAGE-GRPO | 26.0 | 7644 | 3.40 | 36.3 | 5223 | 6.95 |
| $2 \times 10^{-6}$ / $5 \times 10^{-4}$ | GRPO | 23.5 | 8436 | 2.79 | 33.6 | 6491 | 5.18 |
| $2 \times 10^{-6}$ / $5 \times 10^{-4}$ | AdaptThink | 22.0 | 8202 | 2.68 | 32.7 | 4720 | 6.93 |
| $2 \times 10^{-6}$ / $5 \times 10^{-4}$ | ThinkPrune | 19.8 | 6934 | 2.86 | 32.5 | 4790 | 6.78 |
| $2 \times 10^{-6}$ / $2 \times 10^{-3}$ | SAGE-GRPO | 26.2 | 7608 | 3.44 | 36.6 | 5177 | 7.07 |
| $2 \times 10^{-6}$ / $2 \times 10^{-3}$ | GRPO | 23.8 | 8399 | 2.83 | 33.9 | 6452 | 5.25 |
| $2 \times 10^{-6}$ / $2 \times 10^{-3}$ | AdaptThink | 21.8 | 8250 | 2.64 | 33.0 | 4743 | 6.96 |
| $2 \times 10^{-6}$ / $2 \times 10^{-3}$ | ThinkPrune | 19.9 | 6921 | 2.88 | 32.7 | 4802 | 6.81 |

```python
# Initialization and Parameter Processing

class SAGE:
    def __init__(self, llm, tokenizer, eostep_token='\n\n'):
        self.llm = llm
        self.tokenizer = tokenizer
        self._init_special_tokens(eostep_token)

    def forward(self, prompts, params):

        # Process and validate input parameters
        exploration_params = self._process_params(params)

        # Initialize exploration instances for each prompt
        instances = self._init_exploration_instances(prompts, exploration_params)

        # Perform step-wise exploration reasoning
        self._stepwise_exploration(instances, exploration_params)

        # Generate final answers from completed reasoning
        return self._generate_final_answers(instances, exploration_params)

            def _init_special_tokens(self, eostep_token):
        self.eot_token_id = self.tokenizer.convert_tokens_to_ids('</think>')
        self.sot_token_id = self.tokenizer.convert_tokens_to_ids('<think>')
        self.eostep_token_id = self.tokenizer.convert_tokens_to_ids(eostep_token)
        self.eostep_token_list = [self.eostep_token_id, self.eot_token_id]

    def _process_params(self, params):
        processed = {}
        processed['exploration_width'] = params.get('exploration_width', 4)
        processed['max_ans_tokens'] = params.get('max_ans_tokens', 100)
        processed['max_step_num'] = params.get('max_step_num', 10)
        processed['per_step_token'] = params.get('per_step_token', 100)
        processed['enable_sort'] = params.get('enable_sort', True)
        processed['required_completions'] = params.get('required_completions', 1)
        processed['insert_sot'] = params.get('insert_sot', True)
        processed['top_p'] = params.get('top_p', 1.0)
        processed['temperature'] = params.get('temperature', 1.0)
        return processed

    def _init_exploration_instances(self, prompts, params):
        instances = []
        for prompt in prompts:
            # Convert prompt to tokens if it's a string
            prompt_tokens = self.tokenizer.encode(prompt) if isinstance(prompt, str) else prompt

            # Add start-of-thinking token if requested
            if params['insert_sot']:
                prompt_tokens.append(self.sot_token_id)

            # Create exploration instance with empty candidates
            instance = ExplorationInstance(
                prompt_tokens,
                logprobs=None,
                required_completions=params['required_completions']
            )
            instances.append(instance)
        return instances
```

*Figure 15.* Example Python code illustrating the SAGE Initialization and Parameter Processing.

```python
# Step-wise Exploration

    def _stepwise_exploration(self, instances, params):
        for t in range(params['max_step_num']):
            active_instances = [inst for inst in instances if not inst.is_finished]
            if not active_instances:
                break

            for instance in active_instances:
                # Initialize candidates if empty
                if not instance.candidates:
                    self._init_instance_candidates(instance)

                # Generate next reasoning steps
                new_candidates = self._generate_next_steps(instance, params)

                # Process and update candidates
                self._update_instance_candidates(instance, new_candidates, params)

                # Check if instance has enough completions
                if len(instance.completed) >= instance.required_completions:
                    instance.is_finished = True
                    instance.candidates = []

    def _init_instance_candidates(self, instance):
        instance.candidates.append(ExplorationSequence(
            tokens=instance.prompt_tokens,
            logprobs=[],
            cum_logprob=0.0))

    def _generate_next_steps(self, instance, params):
        # Create batch of prompts from current candidates
        candidate_prompts = [cand.tokens for cand in instance.candidates]

        # Configure step sampling parameters
        step_params = SamplingParams(
            n=params['exploration_width'] * 2,
            temperature=params['temperature'],
            top_p=params['top_p'],
            max_tokens=params['per_step_token'],
            stop_token_ids=self.eostep_token_list)

        # Generate next steps using LLM
        outputs = self.llm.generate(
            prompt_token_ids=candidate_prompts,
            sampling_params=step_params,
            use_tqdm=False)

        # Process outputs into new candidates
        return self._process_step_outputs(instance.candidates, outputs)

    def _process_step_outputs(self, current_candidates, outputs):
        new_candidates = []
        for i, (candidate, output) in enumerate(zip(current_candidates, outputs)):
            if output and output.outputs:
                for step_output in output.outputs:
                    new_candidate = ExplorationSequence(
                        tokens=candidate.tokens + step_output.token_ids,
                        logprobs=candidate.logprobs + step_output.logprobs,
                        cum_logprob=candidate.cum_logprob + step_output.cumulative_logprob)
                    new_candidates.append(new_candidate)
        return new_candidates
```

*Figure 16.* Example Python code illustrating the Step-wise Exploration of SAGE.

```python
# Candidate Management and Scoring

    def _update_instance_candidates(self, instance, new_candidates, params):
        completed, ongoing = self._separate_completed_candidates(new_candidates)
        instance.completed.extend(completed)
        if params['enable_sort']:
            ongoing.sort(key=self._score_candidate, reverse=True)
        instance.candidates = ongoing[:params['exploration_width']]

    def _separate_completed_candidates(self, candidates):
        completed = []
        ongoing = []
        for candidate in candidates:
            if candidate.tokens[-1] == self.eot_token_id:
                completed.append(candidate)
            else:
                ongoing.append(candidate)
        return completed, ongoing

    def _score_candidate(self, candidate):
        return candidate.cum_logprob / len(candidate.tokens)

# Final Answer Generation

    def _generate_final_answers(self, instances, params):
        outputs = []
        for instance in instances:
            # Combine and sort all candidates
            all_candidates = instance.completed + instance.candidates
            all_candidates.sort(key=self._score_candidate, reverse=True)

            # Get best candidates and ensure proper formatting
            best_candidates = all_candidates[:instance.required_completions]
            self._finalize_candidates(best_candidates)

            # Generate answers based on completed reasoning
            answer_candidates = self._generate_answers_from_candidates(best_candidates, params)

            # Create output structure
            outputs.append(ExplorationOutput(sequences=answer_candidates))
        return outputs

    def _finalize_candidates(self, candidates):
        for candidate in candidates:
            if self.eot_token_id not in candidate.tokens:
                candidate.tokens.append(self.eot_token_id)

    def _generate_answers_from_candidates(self, candidates, params):
        candidate_prompts = [cand.tokens for cand in candidates]
        answer_params = SamplingParams(
            n=1, temperature=0.0,
            max_tokens=params['max_ans_tokens'])
        answer_outputs = self.llm.generate(
            prompt_token_ids=candidate_prompts,
            sampling_params=answer_params,use_tqdm=False)
        for candidate, output in zip(candidates, answer_outputs):
            if output and output.outputs:
                candidate.tokens += output.outputs[0].token_ids
                candidate.text = self.tokenizer.decode(candidate.tokens)
        return candidates
```

*Figure 17.* Example Python code illustrating the Candidate Management, Scoring and Final Answer Generation of SAGE.

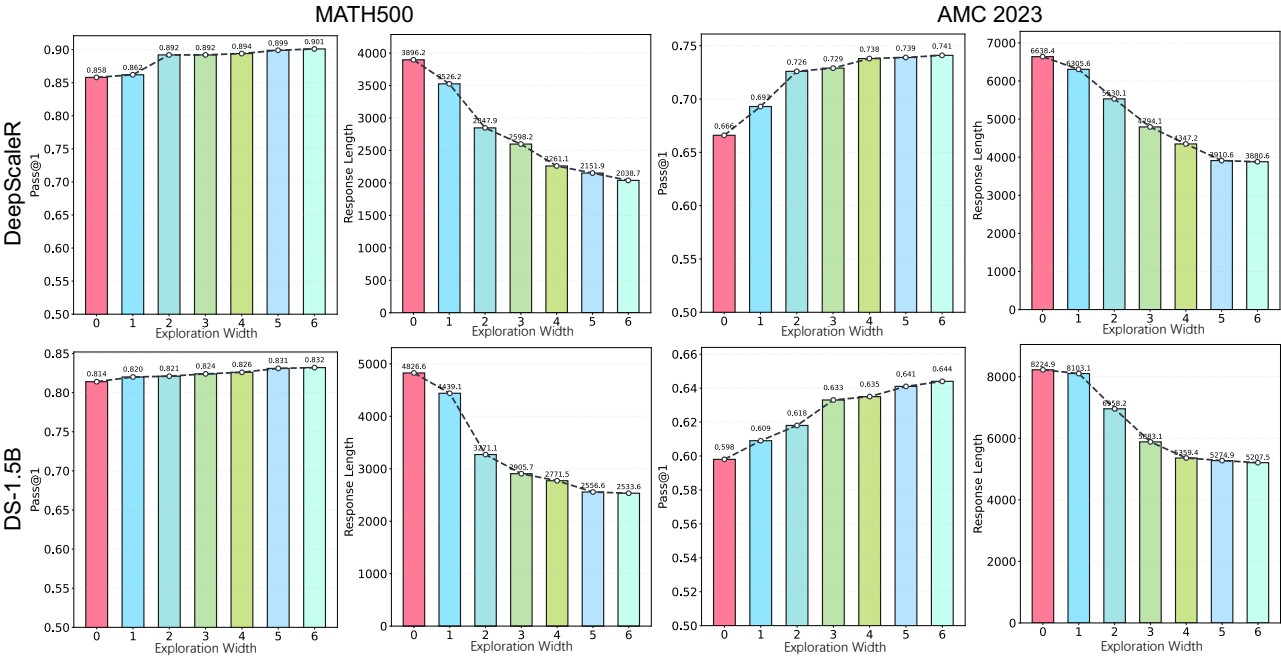

*Figure 18.* Performance on DeepSacleR and DS-1.5B with different exploration width on MATH500 and AMC23. Under all settings, both pass@1 and response length gradually converge.

### DS-1.5B

**Question**                                           Correct Answer: 800

John buys twice as many red ties as blue ties.  The red ties cost 50% more than blue ties. He spent $200 on blue ties that cost $40 each. How much did he spend on ties?

**Reasoning**                                              957 tokens

Okay, so I need to figure out how much John spent on ties. Let's start by parsing the information given step by step.

First, John buys twice as many red ties as blue ties. Hmm, that means if he buys x blue ties, he buys 2x red ties. That makes sense.

Next, the red ties cost 50% more than the blue ties. Alright, so if the blue tie is priced at, say, y dollars, then red ties would cost y plus 50% of y, which is 1.5y. Alternatively, it can be written as 1.5y.

· · ·

Now, let's figure out the cost per red tie. We know that the red ties cost 50% more than the blue ties. The blue tie costs $40, so 50% of 40 is 20. Therefore, each red tie costs 40 + 20 = $60.

Total spent = Total cost of blue ties + Total cost of red ties
Total spent = 200 + 600 = $800.
So, John spent a total of $800 on ties.

Wait, let me double-check my calculations to be sure I didn't make any mistakes.

First, blue ties: 200 / 40 is indeed 5 ties. Red ties are twice that, so 10 ties. Each blue tie is $40, red tie is 50% more, which is $60. 10 red ties at $60 each: 10 * 60 is 600. Blue ties total $200. So 600 + 200 is 800. Yeah, that checks out.

· · ·

Total expenditure: 200 + 600 = 800. Yeah, definitely $800.
I think that even if I messed up in the calculation, this straightforward approach seems correct. So, he spent $800 on ties.

**Final Answer** John spent a total of \boxed{$800} on ties. ✔

### SAGE-GRPO-DS-1.5B

**Question**                                           Correct Answer: 800

John buys twice as many red ties as blue ties.  The red ties cost 50% more than blue ties. He spent $200 on blue ties that cost $40 each. How much did he spend on ties?

**Reasoning**                                              467 tokens

Okay, so I need to figure out how much John spent on ties. Let's start by parsing the information given step by step.

First, John buys twice as many red ties as blue ties, if he buys x blue ties, he buys 2x red ties.

Red ties are 50% more expensive than blue ties. For a blue tie priced at y dollars, the red tie price is 1.5y.

· · ·

Now, let's figure out the cost per red tie. Since red ties are 50% more expensive than blue ties ($40), each red tie costs 40 + 20= $60.

Total spent = Total cost of blue ties + Total cost of red ties
Total spent = 200 + 600 = $800.
So, John spent a total of $800 on ties.

**Final Answer** John spent a total of \boxed{$800 } on ties. ✔

*Figure 19.* Case Study 1

## DS-1.5B

### Question

Correct Answer: (3, \frac{\pi}{2})

Convert the point $(0,3)$ in rectangular coordinates to polar coordinates.

### Reasoning

1381 tokens

Okay, so I need to convert the rectangular coordinate (0, 3) to polar coordinates. Hmm, I remember that polar coordinates are represented as (r, θ)…

First, let me recall the formulas … I think the formula for r … But wait, I also remember that sometimes arctan(y/x) doesn't give the correct angle because it depends on the quadrant where the point is located. So, I need to be careful about that.

Alright, let's apply this to the point (0, 3) … Okay, so r is 3. That makes sense because the point is 3 units above the origin on the y-axis, so the distance from the origin is 3.

. . .

Now, let's figure out θ. Since x is 0, we have θ = arctan(3/0). Wait, division by zero is undefined. Hmm, that's a problem. I need to think about what angle θ would make the point (0, 3) in polar coordinates.

I know that when x is 0, the point lies on the y-axis. If y is positive, it's on the positive y-axis, which is at 90 degrees or π/2 radians from the positive x-axis. If y were negative, it would be at 270 degrees or 3π/2 radians. Since our y is positive, θ should be π/2.

Wait, but sometimes when x is 0, the angle is either π/2 or 3π/2, but since our y is positive, it's π/2. So, θ is π/2.

So, putting it all together, the polar coordinates are (3, π/2). Let me just double-check. … Cos(π/2) is 0, so x is 0. Y is r sin θ = 3 sin(π/2). Sin(π/2) is 1, so y is 3. That gives us (0, 3), which is exactly the point we started with. So, that seems correct.

. . .

So, putting it all together, the polar coordinates are (3, π/2).
 Let me just double-check … That gives us (0, 3),
which is exactly the point we started with. So, that seems correct. 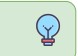

**Final Answer** The polar coordinates are: \boxed{(3, \frac{\pi}{2})} ✓

## SAGE-GRPO-DS-1.5B

### Question

Correct Answer: (3, \frac{\pi}{2})

Convert the point $(0,3)$ in rectangular coordinates to polar coordinates.

### Reasoning

712 tokens

Okay, so I need to convert the rectangular coordinate (0, 3) to polar coordinates. Hmm, I remember that polar coordinates are represented as (r, θ)…

First, let me recall the formulas for converting rectangular coordinates (x, y) to polar coordinates (r, θ). I think the formula for r is the square root of (x squared plus y squared). So, r = √(x² + y²). And for θ, I believe it's the arctangent of y over x, so θ = arctan(y/x).

Alright, let's apply this to the point (0, 3)… Okay, so r is 3. That makes sense because the point is 3 units above the origin on the y-axis, so the distance from the origin is 3.

. . .

I remember that angles in polar coordinates are measured from the positive x-axis, going counterclockwise … Let me confirm that. Yes, π/2 radians is 90 degrees, which points straight up along the y-axis. So, θ should be π/2.

So, putting it all together, the polar coordinates should be (3, π/2).
Let me just write that down: (3, π/2). 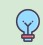

**Final Answer** The polar coordinates are: \boxed{(3, \frac{\pi}{2})} ✓

*Figure 20.* Case Study 2

