# OpenReview forum: "Does Your Reasoning Model Implicitly Know When to Stop Thinking?"
_ICML.cc/2026/Conference — ICML 2026 regular_

### Official Review · Reviewer_uTHs · 2026-02-15

**Soundness:** 2
**Presentation:** 3
**Significance:** 2
**Originality:** 3
**Overall Recommendation:** 4
**Confidence:** 3

**Summary:**

This builds beyond findings that very long and short reasoning traces have lower accuracy: https://arxiv.org/pdf/2505.00127. The authors introduce RFCS, a metric quantifying how early a correct answer appears in a reasoning chain, and show that over half of correct responses contain substantial redundant steps. They then propose TSearch, a beam-search-like algorithm that scores candidate reasoning paths by cumulative average log-probability (Phi) rather than next-token log-probability (phi). The key empirical finding is that Phi-scored paths are both shorter and more accurate, and that the </think> token consistently ranks first in the candidate set when it appears on such paths. This motivates SAGE, a step-level variant, and SAGE-RL, which replaces 2 of 8 rollouts in group-based RL (GRPO/GSPO) with SAGE-discovered chains. Across four models (up to 8B) and six math benchmarks, SAGE-RL improves both accuracy and token efficiency relative to vanilla RLVR and several efficient-reasoning baselines.

Unfortunately, the paper has modest gains on average (they use a lot of datasets and settings), and do hyperparameter tuning of baselines, so the paper should be rejected as is. If the authors add hyperparameter ablations and their method is really still strong, then assuming I trust the results I will likely increase score, potentially to an accept.

**Compliance With Llm Reviewing Policy:**

Affirmed.

**Final Justification:**

Ran comparisons to GRPO and was broadly forthcoming with fixes but didn't tune baselines against other algorithms.

EDIT: comparisons were added. I weakly accept this paper; it needed substantial post-submission changes but ultimately what I asked for was given.

**Key Questions For Authors:**

1) Can you do hyperparmeter sweeps and tuning for baselines to ensure your results are significant (W1)?

2) Are you open to renaming the work given weakness W2?

3) What is training time overhead (W3)?

**Limitations:**

The authors are honest to mention "identical hyperparameter settings for the same base model" but this is a big limitation

**Strengths And Weaknesses:**

### Strengths

**S1. This paper does a great job at turning an interesting finding about which reasoning chains are more accurate into a better algorithm.** The Phi vs phi distinction is the paper's cleanest insight. Cumulative average log-probability selects paths where the model terminates with high confidence (the </think> token ranks first upon appearance), while next-token log-probability does not and leads to length collapse. The ablation via TSearch w/ Phi vs TSearch w/ phi (Figure 4, Table 1) is well-designed and clearly demonstrates this. The convergence analysis (Section 4.3) showing that both accuracy and token efficiency converge as exploration width increases is the strongest evidence that this reflects a genuine property of the model's distribution rather than an artifact of a particular hyperparameter setting.

**S2. SAGE-RL is a practical and elegant integration.** Replacing only 2 of 8 rollouts with SAGE-discovered efficient reasoning chains during RLVR training is simple, requires no reward function modification, and maintains training stability. The fact that this small intervention produces consistent gains across all benchmarks and models is the paper's most practically useful result. At deployment time, the tuned model uses standard pass@1 sampling with no search overhead, which is the right design choice.

**S3. Comprehensive experimental coverage.** Four base models (DS-1.5B, DS-7B, DeepScaleR, Qwen3-8B), six benchmarks, and six baselines. The breadth is commendable and the comparisons are fair within the mathematical reasoning domain. The key result - that competing methods (AdaptThink, ThinkPrune, LC-R1) trade accuracy for compression while SAGE-RL improves both - is consistently demonstrated.

### Weaknesses

**W1. No tuning or even ablation of baseline hparams therefore result significance unclear** The paper studies models with <=8B parameters; while one its own this isn't a problem, since the paper shows very modest accuracy improvements with decent but not crazy length reductions, this brings concerns about undereliciting the baseline: https://safe-lip-9a8.notion.site/Incorrect-Baseline-Evaluations-Call-into-Question-Recent-LLM-RL-Claims-2012f1fbf0ee8094ab8ded1953c15a37

Additionally "identical hyperparameter settings for the same base model" is a bit of a red flag -- see e.g. https://x.com/giffmana/status/2017283232736100742 from a senior researcher in ML (sorry to cite a tweet!). To reproduce:

> never, ever write "we use the same learning rate across all methods for fair comparison"
>
> I read this as "do not trust any of our conclusions" and then i move on.
>
> If learning rate tuning is not mentioned, it takes me a little more time to notice that, but i also move on.

**W2. The central claim is overstated and anthropomorphized.** In the title and throughout the paper, the authors claim that LRMs "implicitly know when to stop thinking." What the evidence actually shows is that a post-hoc search procedure using cumulative log-probability can identify shorter, more accurate reasoning paths among many possible continuations. This is a property of the scoring function applied over the model's probability distribution, not evidence of an internal "knowledge" about stopping. The model doesn't decide to stop and get overruled; rather, among the exponentially many paths the model assigns probability to, some happen to be short, correct, and high-confidence. The greedy/random sampling strategies just don't find them. A more precise framing would be: "LRMs assign high cumulative probability to efficient reasoning paths, but standard decoding strategies fail to select them." The current framing misleads the reader somewhat.

**W3. Training time overhead is not quantified.** The paper's selling point is efficiency, but the cost analysis (Appendix D.4) only covers inference. At inference time the story is clean: SAGE-RL tuned models use standard pass@1 sampling with no search overhead, yielding ~28.7% faster inference on average and >40% latency reduction across most model/benchmark combinations. However, during training SAGE(2,2) generates 2 of 8 rollouts per group via a beam-style search at every step for 600 steps. This must be substantially slower than standard random sampling, but no training wall-clock or FLOP comparison is provided. The paper also does not benchmark inference cost against competing methods (ThinkPrune, AdaptThink, GFPO, etc.); the only inference comparison is SAGE vs Degrade-SAGE. The related work section argues qualitatively that SAGE-RL avoids the repeated rollout costs of S-GRPO/VSRM and the oversampling of GFPO, but this is not measured. For a paper centred on efficiency, the cost analysis needs to cover the full pipeline.

---

> ### Author Rebuttal · Authors · 2026-03-31
>
> We sincerely thank the reviewer for the detailed feedback. We have conducted additional experiments based on your suggestions and hope these responses resolve your concerns.
>
> ---
>
> > W1 & Q1: unclear significance of results
>
> Thanks for raising this concern and provide us with valuable suggestions. We would like to first clarify the rationale behind this choice: since SAGE-RL modifies only the rollout stage of RLVR, we intentionally kept all other hparams identical across methods to ensure a controlled comparison that accurately isolates the improvement attributable to SAGE-RL's design. We also noted that adopting a fixed set of hparams across methods is a common practice in the field of LLM-based RL (e.g., DAPO[1] , Reinforce++[2]).
>
> However, We agree that evaluating across multiple hyperparameter settings would further strengthen the significance of our results. To verify that the effectiveness of SAGE-RL is robust and not an artifact of a particular hyperparameter set, we performed a grid search over the LR and KL coefficient (on DS-1.5B).
>
> | Method     | LR    | KL    |    AIME2025   |  |  OlympiadBench     |  |
> |------------|-------|-------|-------|----------|-------|---------------|
> |            |       |       | pass@1 | LEN      | pass@1 | LEN           |
> | SAGE-GRPO  | 5e-7  | 5e-4  | 25.8  | 7673     | 36.1  | 5253          |
> | GRPO       | 5e-7  | 5e-4  | 23.3  | 8461     | 33.4  | 6536          |
> | SAGE-GRPO  | 5e-7  | 2e-3  | 25.5  | 7804     | 35.8  | 5387          |
> | GRPO       | 5e-7  | 2e-3  | 23.0  | 8598     | 33.1  | 6665          |
> | SAGE-GRPO  | 2e-6  | 5e-4  | 26.0  | 7644     | 36.3  | 5223          |
> | GRPO       | 2e-6  | 5e-4  | 23.5  | 8436     | 33.6  | 6491          |
> | SAGE-GRPO  | 2e-6  | 2e-3  | 26.2  | 7608     | 36.6  | 5177          |
> | GRPO       | 2e-6  | 2e-3  | 23.8  | 8399     | 33.9  | 6452          |
>
> The results demonstrate that the performance gap between SAGE-GRPO and GRPO persists under and even across different hyperparameter combinations, indicating that this superiority stems *not* from hyperparameter tuning, but from the effective transfer of efficient reasoning patterns discovered by SAGE.
>
> ---
>
> > W2 & Q2: overstated core claims
>
> We appreciate that the reviewer has accurately captured the core finding of our work. However, we respectfully disagree with the view that it is merely "a property of the scoring function". We believe this reflects an active capability encoded within the model: if the model's distribution did not systematically encode such a property, no scoring function could reliably discover these efficient reasoning paths.
>
> The $\Phi$ vs. $\phi$ results (Section 4) further supports this: a different scoring function over the same distribution fails entirely, confirming that this property is model-intrinsic rather than scoring-induced.
>
> We are happy to soften the framing such as "LRMs implicitly encode the capability to identify appropriate stopping points" to more precisely convey our finding while avoiding overstatement.
>
> ---
>
> > W3 & Q3: unquantified overhead
>
> We first emphasize that SAGE-RL is only used during training. The inference process during evaluation across all methods and models are identical, with no extra overhead during deployment.
>
> **Theoretical overhead.** The additional cost of SAGE-RL stems from replacing 2 of 8 standard rollouts with SAGE(2,2). SAGE(2,2) keeps 4 candidate sequences, introducing  4× GPU memory overhead for these two trajectories. With sufficient parallelism, the wall-clock time is comparable. However, SAGE's step-by-step inference requires repeated calls to vLLM's generate operator, reducing the efficiency that standard random sampling enjoys. Moreover, SAGE-RL drives faster convergence toward shorter outputs, which reduces rollout latency (Figure 9).
>
> **Empirical results.** We report the overhead results for DS-1.5B. To ablate the impact of vLLM, we also implemented SAGE-GRPO using native HF inference.
>
> | Method      | Inf Engine | Peak GPU Usage | Training Time       |
> |-------------|------------|----------------|---------------------|
> | GRPO        | Verl       | 70089 MiB      | 27.1 h              |
> | SAGE-GRPO   | Verl       | 70091 MiB      | 30.2 h              |
> | GRPO        | HF         | 36118 MiB      | 96.8 h              |
> | SAGE-GRPO   | HF         | 40020 MiB      | 93.2 h              |
> | ThinkPrune  | Verl       | 69991 MiB      | 29.5 h              |
> | LC-R1       | Verl       | 70403 MiB      | 31.8 h   |
> | AdaptThink  | Verl       | 75613 MiB      | 33.8 h              |
>
> SAGE-GRPO has competitive overhead compared with all baselines. With vLLM, it is slightly slower than GRPO but maintains comparable peak GPU usage due to infra constraints. With HF inference, SAGE-GRPO consumes moderately more memory yet runs faster than GRPO, which aligns with our theoretical expectations.
>
> ---
>
> References:
>
> [1] https://arxiv.org/abs/2503.14476
>
> [2] https://arxiv.org/abs/2501.03262v3

---

> > ### Author Rebuttal · Reviewer_uTHs · 2026-04-01
> >
> > Thanks, I updated to a score of 3 but I don't see the tuning of other baselines, e.g. AdaptThink, ThinkPrune, etc. IIUC (i.e. tuning of stuff beyond GRPO)?

---

> > > ### Author Response · Authors · 2026-04-03
> > >
> > > We sincerely thank you for your valuable suggestions and for granting us additional time.
> > > To address your concern, we have conducted hyperparameter ablation experiments for SAGE-GRPO and additional baselines beyond GRPO, including AdaptThink and ThinkPrune-2k.
> > > Specifically, we sweep over four combinations of learning rate (LR ∈ {5e-7, 2e-6}) and KL coefficient (KL ∈ {5e-4, 2e-3}), and train each method independently under every setting. The results are reported below:
> > >
> > >
> > > **LR=5e-7, KL=5e-4:**
> > > | Method     | AIME2025 |        |        | OlympiadBench |        |        |
> > > |------------|----------|--------|--------|---------------|--------|--------|
> > > |            | pass@1   | LEN    | TE($\times 10^{-3}$) | pass@1        | LEN    | TE($\times 10^{-3}$) |
> > > | SAGE-GRPO  | **25.8** | 7673   | **3.36** | **36.1**      | 5253   | **6.87** |
> > > | GRPO       | 23.3     | 8461   | 2.75   | 33.4          | 6536   | 5.11   |
> > > | AdaptThink | 21.9     | 8102   | 2.70   | 32.4          | **4730** | 6.85   |
> > > | ThinkPrune | 19.7     | **6950** | 2.83   | 32.3          | 4793   | 6.74   |
> > >
> > > **LR=5e-7, KL=2e-3:**
> > > | Method     | AIME2025 |        |        | OlympiadBench |        |        |
> > > |------------|----------|--------|--------|---------------|--------|--------|
> > > |            | pass@1   | LEN    | TE($\times 10^{-3}$) | pass@1        | LEN    | TE($\times 10^{-3}$) |
> > > | SAGE-GRPO  | **25.5** | 7804   | **3.27** | **35.8**      | 5387   | **6.65** |
> > > | GRPO       | 23.0     | 8598   | 2.68   | 33.1          | 6665   | 4.97   |
> > > | AdaptThink | 22.0     | 8254   | 2.67   | 31.5          | **4762** | 6.61   |
> > > | ThinkPrune | 19.9     | **6925** | 2.87   | 31.6          | 4779   | 6.61   |
> > >
> > > **LR=2e-6, KL=5e-4:**
> > > | Method     | AIME2025 |        |        | OlympiadBench |        |        |
> > > |------------|----------|--------|--------|---------------|--------|--------|
> > > |            | pass@1   | LEN    | TE($\times 10^{-3}$) | pass@1        | LEN    | TE($\times 10^{-3}$) |
> > > | SAGE-GRPO  | **26.0** | 7644   | **3.40** | **36.3**      | 5223   | **6.95** |
> > > | GRPO       | 23.5     | 8436   | 2.79   | 33.6          | 6491   | 5.18   |
> > > | AdaptThink | 22.0     | 8202   | 2.68   | 32.7          | **4720** | 6.93   |
> > > | ThinkPrune | 19.8     | **6934** | 2.86   | 32.5          | 4790   | 6.78   |
> > >
> > > **LR=2e-6, KL=2e-3:**
> > > | Method     | AIME2025 |        |        | OlympiadBench |        |        |
> > > |------------|----------|--------|--------|---------------|--------|--------|
> > > |            | pass@1   | LEN    | TE($\times 10^{-3}$) | pass@1        | LEN    | TE($\times 10^{-3}$) |
> > > | SAGE-GRPO  | **26.2** | 7608   | **3.44** | **36.6**      | 5177   | **7.07** |
> > > | GRPO       | 23.8     | 8399   | 2.83   | 33.9          | 6452   | 5.25   |
> > > | AdaptThink | 21.8     | 8250   | 2.64   | 33.0          | **4743** | 6.96   |
> > > | ThinkPrune | 19.9     | **6921** | 2.88   | 32.7          | 4802   | 6.81   |
> > >
> > >
> > >
> > > Across all four configurations, the relative ranking of methods is remarkably stable: SAGE-GRPO consistently achieves the highest pass@1 and token efficiency on both benchmarks. AdaptThink and ThinkPrune excel at compression (achieving the shortest response lengths), but this comes at the cost of accuracy — both underperform even vanilla GRPO in pass@1. The consistency of these gaps across independently tuned settings provides evidence that our improvements are not artifacts of a particular hyperparameter choice.
> > >
> > > In the revised version, we will incorporate the aforementioned hyperparameter ablation results in the manuscript for sure.
> > >
> > > We sincerely thank you again for your constructive feedback and hope that these additional experimental results can address your remaining concerns. We would be very grateful if you would consider revising the evaluation in light of these new results.

---

### Official Review · Reviewer_dmEN · 2026-03-10

**Soundness:** 4
**Presentation:** 3
**Significance:** 4
**Originality:** 3
**Overall Recommendation:** 5
**Confidence:** 3

**Summary:**

The paper proposes **Self-Aware Guided Efficient Reasoning** (SAGE), a novel sampling paradigm for Large Reasoning Models. The paradigm utilizes the self-confidence to discover efficient and precise reasoning paths. The paper applies SAGE to GRPO and achieves **higher pass@1** performance with **44% fewer CoT tokens** on math benchmarks like AMC 2023, Math 500, AIME 2025, Minerva, and Olympiad Bench.

**Compliance With Llm Reviewing Policy:**

Affirmed.

**Key Questions For Authors:**

See weaknesses.
Q1. How SAGE works for tool-using and coding tasks?
Q2. Technical novelty compared with DeepConf.
Q3. Comparisons between SAGE and DeepConf under same reasoning budgets.

**Strengths And Weaknesses:**

## Strength

1. Strong experimental results. The proposed SAGE-RL method achieves +2.1% in math benchmarks while reducing the CoT tokens by 44%, which is a great contribution to the LRM efficiency.

2. The claim that "LRM can implicitly know when to stop thinking"  provides good insights to development of LRM. The motivation to use self-confidence is clear and intuitive.

3. The authors conducted comprehensive experiments on Math500, AIME 24/25, and OlympiadBench with various model scopes and scales like Deepseek-1.5B/7B, Qwen3-8B, DeepScaleR, proving the generalization ability of the algorithm.

4. The paper is well written and easy-to-follow, with demo samples and figures for illustration.

## Weakness
1. The author mainly conduct experiments on math reasoning scope. I am curious on how SAGE works for tool-using and coding tasks. Different from math problems which follows a step-by-step reasoning paradigm, tool-using and coding follows a structured output paradigm with long-term relationship.

2. The authors should state the differences and technical novelty between SAGE and DeepConf (Fu, Yichao, et al, "Deep Think with Confidence"). DeepConf is also a sampling algorithm to boost LRM's efficiency and performance at test time. It might be a baseline to compare under the same reasoning budget.

---

> ### Author Rebuttal · Authors · 2026-03-31
>
> Thank you for your valuable feedback.
> We  answered  each of  your detailed questions  and conducted experiments and discussions across a broader set of tasks and baselines as per your suggestions. We hope these responses resolve your concerns, and we are happy to respond any further inquiries you may have.
>
> ---
> >W1 & Q1: Performance on Tool Use and Coding Tasks
>
> We sincerely thank the reviewer for the valuable suggestion to verify the cross-task generalization ability of SAGE. To address this, we have conducted additional experiments on coding tasks using SAGE-RL. Specifically, we used **taco** (12,898 programming problems) as the training set and **lcb_v6** (131 programming problems) as the test set. We fine-tuned DS-1.5B using both SAGE-GRPO and vanilla GRPO. The evaluation results under the same inference settings as Appendix C.2 are summarized in the table below:
>
> | Model      | Pass@1(%) | LEN    | TE($\times 10^{-3}$) |
> |------------|-----------|--------|---------------------|
> | DS-1.5B    | 12.4      | 27659  | 0.448               |
> | +GRPO      | 23.5      | 23847  | 0.985               |
> | +SAGE-GRPO | 27.7      | 12926  | 2.143               |
>
> We were surprised to find that SAGE-GRPO achieves substantially larger gains over the baseline compared to our main mathematical experiments: a 4.2% improvement in pass@1 and more than a twofold reduction in average response length.
>
> Analysis of output examples reveals that this significant improvement stems from the fact that code implementations with identical functionality can vary dramatically in length (e.g., manually implementing an operator versus directly calling it). SAGE effectively selects the shorter implementation branches, yielding token savings that far exceed those obtained from choosing different mathematical reasoning steps.
>
> Since existing work on efficient reasoning has primarily focused on mathematical reasoning, we had overlooked SAGE’s substantial potential on code-related tasks. Due to time constraints, we plan to conduct extensive experiments on more models and code datasets in the revision, and include detailed analysis of the results.
>
> For tool use tasks, we note that SAGE may face unique challenges. Since tool use requires inserting outputs from external tools (e.g., retrieval or Python execution) into the reasoning chain, these out-of-distribution tokens could potentially disrupt the model’s confidence judgment. Therefore, adapting SAGE to tool use represents a promising research direction, which we plan to explore in future work.
>
> ---
>
> > W2 & Q2 & Q3: Comparison with DeepConf
>
> We thank the reviewer for raising this important point. DeepConf is a Test-Time Scaling (TTS) algorithm. For each problem, it sets an outer majority voting threshold and an inner confidence threshold. Generation terminates when the answer consensus exceeds the preset threshold or the maximum number of reasoning traces (512) is reached. If the confidence of the sliding window within a single reasoning trace falls below the threshold, the sequence is discarded early.
>
>
> DeepConf and SAGE both leverage confidence signals to guide reasoning. We summarize the differences between DeepConf and our work as follows.
>
> 1.  DeepConf aims to reduce the computational cost of pass@k by pruning unpromising traces during repeated sampling. SAGE, by contrast, is designed to improve the quality of pass@1 through step-wise search over a broader reasoning space. They target different inference paradigms and address different bottlenecks.
>
> 2. DeepConf relies on externally preset confidence thresholds to decide when to discard a trace. While this shares a surface-level resemblance with our approach, a key contribution of our work (Section 4) is demonstrating that the ability to judge reasoning quality is an inherent capability of the model itself—it does not need to be imposed through external heuristics. SAGE exploits this insight by integrating confidence-aware selection directly into the step-level prefix update, rather than passively truncating sequences after the fact.
>
>
> 3. Additionally, SAGE-RL goes a step further by incorporating SAGE-discovered efficient reasoning patterns into the RL training loop, thereby improving the model's intrinsic reasoning ability. DeepConf operates purely at inference time and does not enhance the model itself.
>
> To provide a fair comparison, we adapt SAGE to the pass@k setting used by DeepConf. Due to time constraints, please refer to the results at the following anonymous link:
> https://anonymous.4open.science/r/icml_rebuttal-E676/
>
> We will discuss DeepConf in the revised manuscript, clearly articulating the shared motivation and the distinctions outlined above.

---

> > ### Author Rebuttal · Reviewer_dmEN · 2026-04-04
> >
> > I have no more questions.

---

> > > ### Author Response · Authors · 2026-04-06
> > >
> > > Dear Reviewer dmEN,
> > >
> > > Thank you for acknowledging that our responses have fully addressed your concerns and for your positive evaluation of SAGE's contribution to the LLM reasoning community. Your thoughtful and constructive feedback has been immensely valuable in enhancing our work. We will be sure to incorporate all of your suggestions and discussions into the revised manuscript.
> > >
> > > Best regards,
> > >
> > > Authors

---

### Official Review · Reviewer_hxML · 2026-03-15

**Soundness:** 3
**Presentation:** 1
**Significance:** 2
**Originality:** 2
**Overall Recommendation:** 4
**Confidence:** 3

**Summary:**

This paper investigates whether LRMs implicitly know when to stop thinking. The authors introduce SAGE (Self-Aware Guided Efficient Reasoning), a sampling paradigm that leverages average cumulative log-probability to identify shorter, more effective reasoning chains. They further propose SAGE-RL, which integrates SAGE into the rollout process of RLVR. Experiments across six mathematical benchmarks show improvements in both accuracy and token efficiency.

**Compliance With Llm Reviewing Policy:**

Affirmed.

**Ethical Review Concerns:**

No ethical concerns.

**Final Justification:**

I increased Soundness and Overall Recommendation after reading the authors' response.

**Key Questions For Authors:**

- Q1. The key motivation for SAGE—that reasoning models implicitly know when to stop thinking—is primarily supported by experiments on MATH-500 (with occasional reference to AMC23). Have the authors verified whether the same phenomenon holds on additional reasoning datasets (e.g., AIME, OlympiadBench, or other math benchmarks)?
- Q2. SAGE replaces token-level expansion with step-wise reasoning chain expansion, but the paper does not clearly specify how a reasoning step is defined or segmented in practice. How are reasoning steps identified (e.g., newline delimiters, special tokens, or model-generated boundaries), and how sensitive is SAGE to this choice?
- Q3. The paper motivates step-wise expansion using observations from TSearch w/ Φ, but token-level and step-level search spaces may behave differently in terms of exploration and pruning. Have the authors conducted experiments isolating the effect of step-wise expansion itself (e.g., comparing token-level vs step-level exploration under similar budgets)?
- Q4. The ablation studies evaluate exploration width only within a relatively small range (e.g., ≤6). Have the authors explored larger widths (e.g., 20–50) or different step budgets to better characterize scaling behavior and the robustness of the stopping condition?
- Q5. Clarification of training and evaluation protocols. Some aspects of the training and evaluation setup are unclear. Specifically:
    - How was the 2 SAGE + 6 random trajectories mixture chosen during RL training?
    - Why is the training length 9k, which is shorter than related work such as DAPO paper (20k)?
    - During evaluation in Table 2, is SAGE used for inference, or only during training?
    - Do all baselines use the same inference sampling strategy in Table 2?

**Limitations:**

The paper does not include a Limitations section. It should acknowledge that the analysis and evaluation focus primarily on mathematical reasoning benchmarks. It remains unclear whether the observed stopping behavior and the proposed SAGE sampling strategy generalize to other reasoning domains, such as general science reasoning or coding.

**Strengths And Weaknesses:**

## Strengths

* **Interesting observation about reasoning termination.**
  The empirical analysis suggesting that models may implicitly recognize when to stop reasoning is a compelling insight that could have broader implications for inference-time scaling methods.

* **Practical efficiency motivation.**
  Reducing unnecessary reasoning tokens while maintaining or improving accuracy is an important problem for deploying large reasoning models.

---

## Weaknesses

### 1. Limited validation of the core analytical observations

The key motivation for SAGE relies heavily on experiments conducted on **MATH-500** (with occasional use of AMC23). Since these observations form the conceptual basis of the method, it would strengthen the paper to verify whether the same behavior holds across a broader set of reasoning datasets (e.g., AIME, OlympiadBench, or other reasoning benchmarks).

### 2. The transition from token-level search to step-wise expansion needs stronger justification

The paper argues that because the `</think>` token consistently ranks highly under TSearch w/ Φ, token-level expansion can be replaced with **step-wise reasoning expansion**. However, the connection between these two formulations is not fully established.

In particular:

* The definition of a **“reasoning step”** is unclear (e.g., how steps are segmented).
* Token-level and step-level search spaces may behave differently in terms of exploration and pruning. Further experiments are needed to support this transition.

A more detailed analysis or ablation isolating this design choice would strengthen the argument.

### 3. Ablation studies are somewhat limited

Some ablations explore relatively narrow parameter ranges. For example, exploration width appears to be tested only up to small values (e.g., ≤6). For a search-based method, evaluating larger widths (e.g., 20–50) would better characterize scaling behavior and robustness of the termination strategy.

Similarly, the interaction between exploration width and the stopping condition could be analyzed more thoroughly.

### 4. Several training design choices lack justification

The RL training setup mixes trajectories generated via SAGE and random sampling (e.g., 2 vs 6 trajectories), but the rationale behind this distribution is not clearly explained.

Additionally, the training schedule differs from related work (e.g., shorter training length (9k) compared to DAPO (20k)), and it is unclear whether this choice affects the reported results.


### 5. Evaluation setup is sometimes unclear

Some aspects of the evaluation protocol are difficult to interpret:

* In Section 5.2, **SAGE (2,1)** suggests only one trajectory is retained, but the reported metrics on MATH-500 and AMC23 appear to use **multiple samples (mean@4 or mean@16)**.
* In Table 2, it is not always clear whether **SAGE is used during training only, or both training and inference**.
* It would also help to clarify whether all baselines use the **same inference sampling strategy**, which is important for fair comparison.

---

> ### Author Rebuttal · Authors · 2026-03-31
>
> We thank the reviewer for the thorough and constructive feedback. We recognize that several aspects of our presentation lacked sufficient clarity and appreciate the opportunity to address them.
>
> ---
> > W1 & Q1: Motivation generalizability
>
> We analyze the rank of eot token (end of thinking) in vocabulary probabilities via TSearch with $\Phi$ on AIME and OlympiadBench using DS-1.5B and DeepScaleR. Results show its average rank remains first, validating our claim in Section 4. We further test SAGE on broader benchmarks:
>
> |         Model         |    Method            | OlympiadBench | | AIME 2024 |  |
> |------------------|----------------|---------------|---------------|-----------|-----------|
> |          |       | pass@1        | LEN           | pass@1    | LEN       |
> | DS-1.5B          | SAGE (2,1)     | 38.7          | 4833          | 30.3      | 6840      |
> | | Degrade-SAGE   | 33.8          | 8942          | 26.1      | 12537     |
> | DeepScaleR       | SAGE (2,1)     | 39.1          | 4211          | 36.4      | 7798      |
> |       | Degrade-SAGE   | 36.1          | 6053          | 31.7      | 9444      |
>
> SAGE achieves clear pass@1 gains and shorter average response lengths, further verifying our method's generalizability.
>
> ---
>
> > W2 & Q2 & Q3: Unclear step partitioning, transition from token-wise to step-wise search.
>
> We apologize for any confusion. We split reasoning steps by "\n\n" (Section 2, Line 90) and use this setting by default in later sections.
>
> As steps serve as the logical reasoning units in LRMs, token-wise and step-wise search behave highly consistently. We verify this via the following analyses.
>
> **1. Termination Behavior**: In autoregressive inference, eot token appears at the end of a reasoning step, not an arbitrary token. Thus, they are equivalent for searching reasoning termination signals.
>
> **2. Intermediate behavior**: We sample and mix 100 problems from AIME24, MATH-500, and AMC-23. For each problem, we use DS-1.5B to randomly sample 2–4 reasoning steps to form complete-step prefixes. As a control, we generate random 512–1024 token responses to obtain random-position prefixes. We then run TSearch (2,2) on these prefixes to infer one step, encode all TSearch candidates via all-MiniLM-L6-v2, and compute cosine similarity.
>
> | Prefix    | Similarity |
> |----------------------|--------------------|
> | Complete Steps       | 0.6883      |
> | Random Positions     | 0.8897   |
>
> When TSearch is performed inside reasoning steps (Random Positions), the similarity among different branches is substantially higher than when starting from complete steps. **This eliminates the need for conducting token-level search within reasoning steps**.
>
> **3. Overall performance**: We adopt the settings in Table 1 and report the performance of SAGE.
>
> | Method | TR    | ACC   | T-LEN | LEN   |
> |----------------------|-------|-------|-------|-------|
> | TSearch (4,1) w/ $\Phi$ | 1.00  | 0.92  | 2213  | 2609  |
> | SAGE(4,1)            | -     | 0.92  | 2234  | 2655  |
>
> As shown in the table, the performance of SAGE(4,1) is fully aligned with that of TSearch, indicating highly consistent behavior between them.
>
> In summary, step-wise search behaves consistently with token-wise search, while lowering overhead by reducing candidate sequence maintenance. Thus, switching from TSearch to SAGE is natural and reasonable.
>
> ---
>
> > W3 & Q4: narrow exploration width (EW) and step budget range
>
> TSearch performance converges clearly as EW increases from 1 to 6 (Figure 15), making a maximum EW of 6 reasonable. Additionally, vLLM only supports up to top-16 (when EW=6, we keep top-12) logprobs per token, indicating little value in keeping extremely low-rank tokens.
>
> We have analyzed the scaling trend of SAGE across steps from 10 to 80 (Section 5.2).
>
> ---
>
> > W4: Lack of rationality in RL training settings
>
> A group size of 8 is a common choice in the LLM-RL field, such as in DeepSeekMath and LC-R1.
>
> As shown in Figure 14, latency overhead rises sharply when EW exceeds 2, while Figure 15 demonstrates marginal performance gains from further increasing EW. Thus, $EW=2$ achieves the optimal cost-performance trade-off. Under this setting, performance differences between 2 or 1 SAGE trajectories are negligible, as stated in Appendix D.2.
>
> For training efficiency, we cannot use an extremely long context while keeping other parameters reasonable. Meanwhile, we apply the same token budget to all baselines to ensure fair comparison.
>
> ---
>
> > W5 & Q5: Occasionally unclear evaluation settings
>
> mean@4 and mean@16 represent the average pass@1 over repeated runs to reduce evaluation randomness. We apologize for the ambiguity and will update the description to "average pass@1 over multiple repeated experiments" in the revised paper.
>
> **SAGE is only used in training, while all models employ identical standard random sampling during evaluation** (Appendix C.2). We apologize for the confusion and will highlight this in Section 6 of the revised manuscript.

---

> > ### Author Rebuttal · Reviewer_hxML · 2026-04-04
> >
> > Thanks for the author response, my concerns have been resolved, I will raise my score.

---

> > > ### Author Response · Authors · 2026-04-06
> > >
> > > Dear Reviewer hxML,
> > >
> > > Thank you very much for acknowledging that our responses have fully resolved your concerns, and for your positive assessment of SAGE's contribution to the LLM reasoning community. Your detailed and constructive feedback has been invaluable in strengthening our work. We will definitely incorporate all of your suggestions and discussions into the revised manuscript.
> > >
> > > Best regards,
> > >
> > > Authors

---

### Decision · Program_Chairs · 2026-04-30

**Decision:**

Accept (regular)

**Comment:**

This work makes the observation that LRMs may know when to terminate reasoning implicitly. Then, it proposes a new sampling paradigm SAGE that leverages this to improve performance while reducing token consumption.